

# Validation of a new global irrigation scheme in the land surface model ORCHIDEE v2.2

Pedro F. Arboleda-Obando[1], Agnès Ducharne[1], Zun Yin[2], and Philippe Ciais[3]

[1]Laboratoire METIS (UMR 7619), IPSL, Sorbonne Université, CNRS, EPHE, Paris, 75005, France
[2]Program in Atmospheric and Oceanic Sciences, Princeton University, Princeton, 08540, New Jersey, USA
[3]Laboratoire des Sciences du Climat et de l'Environnement, IPSL, CNRS-CEA-UVSQ, Gif-sur-Yvette, 91191, Essonne, France

**Correspondence:** Pedro F. Arboleda-Obando (pedro.arboleda_obando@upmc.fr)

**Abstract.**

Irrigation activities are important for sustaining food production, and account for 70% of total global water withdrawals. In addition, due to increased evapotranspiration (ET) and changes on leaf area index (LAI), these activities have an impact on hydrology and climate. In this paper we present a new irrigation scheme within the land surface model ORCHIDEE. It
5   restrains actual irrigation according to available freshwater by including a simple environmental limit and using allocation rules depending on local infrastructure. We perform a simple sensitivity analysis and parameter tuning to set the parameter values and match the observed irrigation amounts against reported values, assuming uniform parameter values over land. Our scheme matches irrigation withdrawals amounts at global scale, but we identify some areas in India, China and the US (some of the most intensively irrigated regions worldwide) where irrigation is underestimated. In all irrigated areas, the scheme
10   reduces the negative bias of ET. It also exacerbates the positive bias of the leaf area index (LAI) except for the very intensively irrigated areas, where irrigation reduces a negative LAI bias. The increase of ET decreases river discharge values, in some cases significantly, although this does not necessarily lead to a better representation of discharge dynamics. Irrigation, however, does not have a large impact on the simulated total water storage anomalies (TWSA) and its trends. This may be partly explained by the absence of non-renewable groundwater use, and its inclusion could increase irrigation estimates in arid and semiarid
15   regions by increasing the supply. Correlation of irrigation biases with landscape descriptors suggests that inclusion of irrigated rice and dam management could improve the irrigation estimates as well. Regardless of this complexity, our results show that the new irrigation scheme helps simulating acceptable land surface conditions and fluxes in irrigated areas, which is important to explore the joint evolution of climate, water resources and irrigation activities.





## 1 Introduction

Irrigation seeks to increase crop yields by reducing plant water stress (Siebert and Döll, 2010; Klein Goldewijk et al., 2017), and supports about 43% of the world's food production on about 20% of arable land (Siebert and Döll, 2010; Grafton et al., 2017). The beneficial effects of irrigation on food production, population and economic growth, have dramatically pushed the increase of irrigated areas during the 20th century, from 28 Mha in 1850 to 276 Mha in 2000 (Klein Goldewijk et al., 2017; Siebert et al., 2015). As a consequence, by the year 2000, irrigation accounted for 70% of the total water withdrawn (between 2657 and 3594 km³/year). The consumptive water use, i.e. the part of the withdrawn water that actually becomes evapotranspiration (ET) and does not flows to surface supplies and groundwater, represents half of that volume (between 1021-1598 km³/year, around 1.7% of total continental ET of 75.6x10³ km³/year according to Jung et al. (2019)) and represents around 90% of the total consumptive water use by human activities (Pokhrel et al., 2016; Döll et al., 2012; Hoogeveen et al., 2015).

Water abstraction and corresponding ET increase have a direct impact on the water and energy balances, and on surface and subsurface hydrology (Döll et al., 2012; Taylor et al., 2013; Vicente-Serrano et al., 2019). Atmosphere reacts as well to these changes on land surface fluxes, for instance with regional increases/decreases in rainfall rate, or decreases in temperature extremes (Lo and Famiglietti, 2013; Guimberteau et al., 2012b; Cook et al., 2015; Al-Yaari et al., 2019; Thiery et al., 2020). Thus it was recently shown that climate models better capture historical trends in evapotranspiration if they account for irrigation and its expansion, although the resulting cooling effect is too strong if irrigation is not limited by water availability (Al-Yaari et al., 2022). Finally, with the acceleration of climate change, the irrigation water demand is likely to increase, not only by expansion of the irrigated area, but also by increasing temperature and changing precipitation variability (Wada et al., 2013). All these impacts and effects have promoted the inclusion of irrigation inside land surface models (LSMs), which represent the continental branch of the hydrologic cycle in the earth system models (Pokhrel et al., 2016).

Besides LSMs, global hydrology models (GHMs) also represent irrigation at global scale. Originally, GHMs were developed to assess water resources availability and water use. In GHMs, irrigation demand is equal to the increase of ET due to irrigation (i.e. water that becomes evapotranspiration). This ET increase is estimated as the differences between crop-specific potential evapotranspiration PET, $ET_0$, and actual ET with no irrigation (Siebert and Döll, 2010; Mekonnen and Hoekstra, 2011; Wada and Bierkens, 2014; Hanasaki et al., 2018; Chiarelli et al., 2020). This reference PET $ET_0 = k_c \cdot$ PET, where $k_c$ is a crop-type and growing stages dependent parameter. These models also consider delivery losses and flows that return to natural reservoirs, i.e. they consider the total water withdrawal, by using empirical ratios or specific rules according to the irrigation methods (Rost et al., 2008; Jägermeyr et al., 2015). The advantage of calculating the withdrawn volume is that it allows comparison and validation with datasets of reported values, for example the FAO AQUASTAT dataset (Frenken and Gillet, 2012). GHMs explicitly represents water supply sources (Döll et al., 2012) and allows the estimation of non-sustainable groundwater used for irrigation (Wada et al., 2012). Some GHMs also simulate water allocation (use of water by type of source) based on rules that use information of local infrastructure and environmental flow estimations (Siebert et al., 2010; Hanasaki et al., 2008a).

LSMs do not use in general a PET to estimate irrigation demand. The reason is that LSMs do not deduce ET from daily PET input data but from surface energy balance at hourly and subhourly time steps. This difference rises consistency issues





between empirical PET formulas and potential ET rates in LSMs (Barella-Ortiz et al., 2013). Some LSMs prescribe irrigation
rates estimated offline (Lo and Famiglietti, 2013; Cook et al., 2015), but most of LSMs and some GHMs estimate irrigation
demand by calculating a deficit, for instance, a soil moisture deficit between actual and a target soil moisture (Haddeland et al.,
2006; Hanasaki et al., 2008a; Leng et al., 2014; Pokhrel et al., 2015; Jägermeyr et al., 2015). Note that the deficit approach
includes flows that return to natural reservoirs according to the represented irrigation technique, but delivery losses are not
explicitly included (Yin et al., 2020; Leng et al., 2017). In addition, irrigation shortage due to water availability is not well
represented in LSMs, as some of them include a virtual infinite reservoir to fulfill irrigation demand (Ozdogan et al., 2010;
Leng et al., 2014; Pokhrel et al., 2012). This virtual reservoir may represents fossil groundwater use and water table depletion,
which is important in some areas like US high plains and India (Pokhrel et al., 2015; Leng et al., 2017; Felfelani et al., 2021).
Water allocation is commonly based on a stream-water-supply first rule (Guimberteau et al., 2012b) with some exceptions that
use the global groundwater inventory from Siebert et al. (2010) (Leng et al., 2017; Felfelani et al., 2021). These rather simple
irrigation schemes are used in land surface-atmosphere simulations to assess irrigation effects on climate (Puma and Cook,
2010; Lo and Famiglietti, 2013; Guimberteau et al., 2012b; Lo et al., 2021) but not on water resources assessment.

ORCHIDEE, the LSM of the IPSL (Institut Pierre Simon Laplace) Earth system model (Krinner et al., 2005; Boucher
et al., 2020) has been used to assess irrigation effects on climate. First attemps to crudely represent irrigation were based on
potential evaporation and potential transpiration for a generic crop type (de Rosnay et al., 2003; Guimberteau et al., 2012b). This
irrigation scheme restraints irrigation according to available water, and includes simple allocation rules. Recently, ORCHIDEE-
CROP, a version of the model that includes a crop phenology module, improved the irrigation scheme by representing flood
and paddy irrigation techniques, and was tested in offline mode in China (Yin et al., 2020). Latest improvements open the
possibility to assess irrigation effects on water resources. This is important, as there is evidence that some modelling biases
within ORCHIDEE in offline and coupled modes are correlated to the surface equipped for irrigation (Mizuochi et al., 2021).
Here, we present evidence on the effect of irrigation on reduction of modelling biases in some key variables like ET and leaf
area index (LAI), and on river discharge and total water storage dynamics (TWSA). After describing the model and the global
irrigation scheme, we set the parameter values by using short simulations. We perform a sensitivity analysis and a simple
parameter tuning to fit to observed irrigation rates. We then perform long simulations and we compare irrigation estimates
to observations and corresponding variability due to parameter values and input maps. We validate irrigation estimates by
reported values, and we asses the spatial variability of the modelling bias. Then we assess the modelling bias against observed
datasets using a factor analysis, with and without irrigation, for ET and LAI. In large basin with extensive irrigation activities,
we compare simulated and observed values of discharge, and total water storage anomalies (TWSA). We also show some
results on the correlation between the irrigation bias and some landscape descriptors, as a first step to improve the realism of
the scheme. Finally, we discuss the results and we present the main conclusions and perspectives.



## 2 Model description

### 2.1 ORCHIDEE v2.2

ORCHIDEE describes the fluxes of mass, momentum, and heat between the surface and the atmosphere (Krinner et al., 2005). Here we use version 2.2, which is close to the version used for CMIP6 (corresponding to 2.0). Version 2.0 has been largely described in many papers (Cheruy et al., 2020; Boucher et al., 2020; Tafasca et al., 2020) and version 2.2 only adds a few minor bug corrections. We summarize the main characteristics of the model that mediate in the simulation of irrigation.

In each grid cell, vegetation is represented by a mosaic of up to 15 plant functional types (PFTs), including generic C3 and C4 crops, as well as generic C4 grasses, and tropical, boreal and temperate C3 grasses. The PFTs fractions are described by the LUHv2 dataset (Lurton et al., 2020), and each PFT is characterized by a specific set of parameters, applied to same set of equations (Boucher et al., 2020; Mizuochi et al., 2021). Plant phenology is controlled by the STOMATE module, which couples photosynthesis and the carbon cycle and computes the evolution of the leaf area index (LAI), all these processes depending on $CO_2$ atmospheric concentration (Krinner et al., 2005).

A specialized version of ORCHIDEE has been proposed by Wu et al. (2016) and evaluated by Müller et al. (2017) to better describe temperate crops, with phenology thresholds based on accumulated degree days after sowing date, improved carbon allocation to reconcile the calculations for leaf and root biomass and grain yield, and nitrogen limitation related to fertilization. It was not used in this work by lack of ubiquitous parameters at global scale, so that C3 and C4 crops are simply assumed to have the same phenology as natural grasslands, but with higher carboxylation rates and adapted maximum possible LAI (Krinner et al., 2005).

Roots constitute an important link between the carbon and the water balance. In each PFT, root density decreases exponentially with depth, and the parameter that controls the decay is PFT-dependent. It is worth noting that the root density profile is constant in time and goes down to the bottom of the soil column, set at 2 m, but forest PFTs have much denser roots than crop and grass PFTs, especially in the bottom part of the soil (Wang et al., 2018). The resulting root density profile is combined with the soil moisture profile and a water stress function to define the water stress factor of each PFT on transpiration (Tafasca et al., 2020) and to estimate the water uptake for transpiration (de Rosnay et al., 2002).

Evapotranspiration is represented by a classical aerodynamic approach and is composed of snow sublimation, interception loss, bare soil evaporation (E), and transpiration (T). The first two proceed at a potential rate, while bare soil evaporation is limited by upward diffusion of water through the soil, and transpiration is controlled by a stomatal resistance, which depends on soil moisture and vegetation parameters. The vegetation types are grouped into three soil columns according to their physiological behavior: high vegetation (eight forest PFTs), low vegetation (six PFTs for grasses and crops), and bare soil. While the energy balance is calculated for the whole grid cell (Boucher et al., 2020), a separate water budget is calculated independently for each soil column, in order to prevent forest PFTs from depriving the other PFTs of soil moisture.

Vertical soil water flow is represented by a 1-D Richards equation coupled to a mass balance, and lateral flow between cells and soil columns is neglected (de Rosnay et al., 2002; Campoy et al., 2013). Here, soil depth is set to 2 meters, and discretized into 22 layers here to finely model lower layers implicated in drainage. Infiltration is simulated as a sharp wetting front based



on the Green and Ampt model (Tafasca et al., 2020; D'Orgeval et al., 2008). The resulting increase in top soil moisture is redistributed by the Richards equation. The bottom boundary condition assumes free drainage, equal to the hydraulic conductivity of the deepest node. The saturated hydraulic conductivity decreases with depth, but roots increase the hydraulic conductivity near the surface (D'Orgeval et al., 2008). Soil parameters are a function of soil texture (Tafasca et al., 2020), and the spatial distribution is taken from the Zobler (1986) map.

A routing scheme transfers surface runoff and drainage from land to the ocean through a cascade of linear reservoirs (Ngo-Duc et al., 2007; Guimberteau et al., 2012a). Each grid cell is split into sub-basins according to a 0.5° flow direction map. Tree reservoirs are considered inside every sub-basin, representing groundwater, overland, and river reservoir, and each one presents a distinct residence time (Ngo-Duc et al., 2007). The groundwater reservoir collects drainage from the soil column, while the overland reservoir collects surface runoff. Both reservoirs are internal to each subbasin and flow to the stream reservoir, which also collects streamflow from the upstream basins and contributes to large-scale routing across subbasins and grid cells. Note that there are two surface reservoirs, overland representing the headwater streams, and river reservoir representing large rivers.

The water and energy budgets and the routing scheme are computed at the same 30-minute time step, while the carbon and plant phenology processes in STOMATE are solved with a daily time step.

## 2.2 Irrigation scheme

The irrigation scheme (Figure 1) is based on the flood irrigation representation from Yin et al. (2020), but it includes some changes in the parameterization to run at global scale. The flood irrigation technique (which consists of adding water to the soil surface to achieve a certain soil moisture content) is chosen for global simulations, as it is the most used (Jägermeyr et al., 2015; Sacks et al., 2009).

Firstly, the scheme defines a root zone depth in the crop- grass soil column, based on the cumulative root density (CRD), ranging from 0 at the soil surface to 1 at the soil bottom: the root zone comprises all soil layers with a CRD below a user-defined threshold, $\text{Root}_{\text{lim}}$. When the threshold is set to 0.9, the root zone includes 90% of the root system. For a 2-m soil column with 22 layers, and an exponential root density decay of 4 (default value for crops and grasses in ORCHIDEE), this threshold defines a root zone depth of 0.5 m, encompassing 10 soil layers.

We can then define a soil moisture deficit $D$ [mm] in the root zone, as the sum of the difference between actual soil moisture and a soil moisture target, in all layers of the root zone:

$$D = \sum_{i \in \text{Root zone}} \max(0, \beta W_i^{fc} - W_i), \tag{1}$$

where $W_i$ and $W_i^{fc}$ (both in mm) are the actual and field capacity soil moisture in soil layer $i$, respectively, and $\beta$ is a user-dependent parameter that controls the target value. To prevent irrigation when there is not plant development, for example during winter, we set the deficit $D$ to zero if all crops and grasses are below a certain LAI threshold, $\text{LAI}_{\text{lim}}$.

The irrigation requirement $I_{\text{req}}$ [mm/s] is calculated as:

$$I_{\text{req}} = f_{\text{irr}} \min(D/\Delta t, I_{\text{max}}), \tag{2}$$





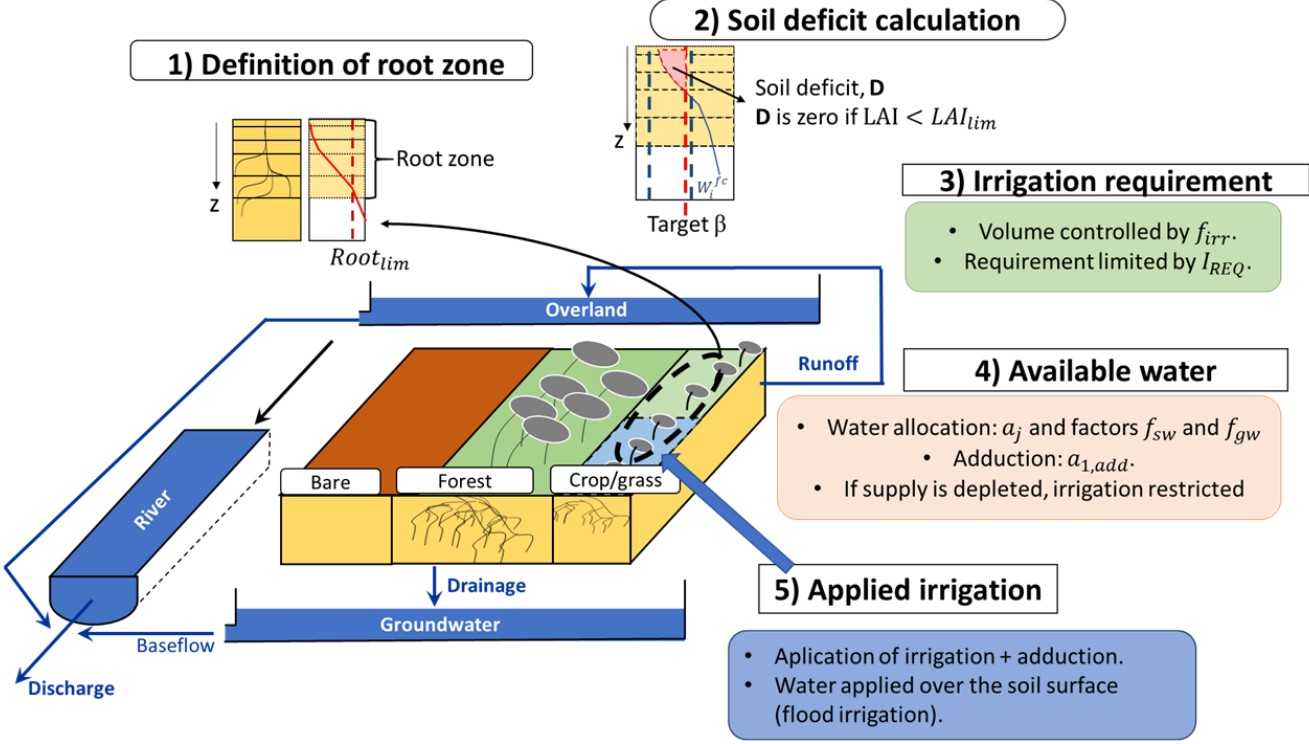

**Figure 1.** ORCHIDEE model and new irrigation scheme. See text for explanation of parameters.

where $f_{irr}$ is the fraction of irrigated surface [-] in the grid cell, defined by a map of irrigated fractions. The map that prescribes the irrigated fraction may change every year, but note that we do not separate the irrigated area into a separate soil column, i.e. the soil column includes crops (both irrigated and rainfed) and grasses. $I_{max}$ is a user-defined maximum hourly irrigation rate [mm/h]. This threshold is used to avoid excessive runoff production when the deficit is large, because the scheme

assumes that the requirement will be fulfilled in the next time step, and the actual irrigation rate applied to the soil surface could exceed the infiltration capacity of the soil. The effective irrigation ($I$, see below) is uniformly applied over the crops and grasses soil column. Therefore, care must be taken by the model that the fraction irrigated is not greater than the crop and grasses soil column. In cases when the fraction of irrigated area is smaller than the fraction of soil column, irrigation will eventually be applied over a larger area than the actually irrigated surface. The difference in size could result in a larger fraction

of irrigation water actually evaporating than in reality, and could lead to an overestimation of the evapotranspiration increase, especially in areas that are energy-controlled (Puma and Cook, 2010).

Irrigation can be withdrawn from three routing reservoirs, but the effective water availability, $A_w$ [mm], also depends on the facility to access surface water and groundwater, and it can be reduced to preserve environmental flows:

$$A_w = f_{sw}(a_1 S_1 + a_2 S_2) + f_{gw} a_3 S_3. \tag{3}$$





In this equation, $S_j$ [mm] is the volume storage in each routing reservoir, with index $j$ equal 1, 2 and 3 for the stream, overland, and renewable-groundwater reservoirs, respectively. To prevent from complete depletion of these reservoirs, which all feed streamflow and support aquatic ecosystems, we mimic an environmental flow regulation by reducing the available volume owing to a user-defined parameter $a_j$, between 0 and 1. It is set here to 0.9 for all three reservoirs, such as to keep at least 10% of the available water at each time step. The facility to irrigate from surface water reservoirs ($S_1$ and $S_2$) and

groundwater reservoir ($S_3$) is accounted for by factors $f_{sw}$ and $f_{gw}$, also ranging between 0 if the reservoirs cannot be used and 1 if they are fully accessible. In the present application, these factors represent the fraction of irrigated areas that are equipped for irrigation with surface and ground water, respectively, following the global map of Siebert et al. (2010). We do not consider irrigation from non-conventional sources (e.g., waste water and water from desalinization plants). This map assumes that a grid-cell is either equipped for groundwater irrigation or for surface water irrigation, so $f_{sw} + f_{gw} = 1$.

Eventually, the actual irrigation $I$ [mm/s] is estimated at each time step by comparing $I_{\mathrm{req}}$, i.e. the demand, to water availability $A_w$ [mm], i.e. the supply:

$$I = \min(A_w/dt, I_{\mathrm{req}}). \tag{4}$$

If we assumed that water abstraction $Q_j$ from each natural reservoir due to irrigation withdrawal is simply proportional to available water in each of them, it would be given by the following equations, the sum of the three right-hand side terms being

equal to $I$:

$$\frac{dS_1}{dt} = -Q_1 = -\frac{f_{sw}\, a_1\, S_1}{A_w} I \tag{5}$$

$$\frac{dS_2}{dt} = -Q_2 = -\frac{f_{sw}\, a_2\, S_2}{A_w} I \tag{6}$$

$$\frac{dS_3}{dt} = -Q_3 = -\frac{f_{gw}\, a_3\, S_3}{A_w} I \tag{7}$$

But we chose to implement an additional constraint for surface water withdrawals, which are withdrawn from the stream

reservoir (corresponding to large rivers) in priority. This new constraint leads to define the revised set of equations, where the total surface water availability is $A_{sw} = f_{sw}(a_1 S_1 + a_2 S_2)$:

$$\frac{dS_1}{dt} = -Q_1 = -\min\left(\frac{A_{sw}}{A_w} I, \frac{f_{sw}\, a_1\, S_1}{\Delta t}\right) \tag{8}$$

$$\frac{dS_2}{dt} = -Q_2 = -\min\left(\frac{A_{sw}}{A_w} I - Q_1, \frac{f_{sw}\, a_2\, S_2}{\Delta t}\right) \tag{9}$$

$$\frac{dS_3}{dt} = -Q_3 = -\frac{f_{gw}\, a_3\, S_3}{A_w} I \tag{10}$$

The sum of $Q_1$, $Q_2$, and $Q_3$, still equals $I$.

     When $I_{\mathrm{req}} - I > 0$, i.e. there is a deficit and the water supply cannot satisfy the irrigation demand, the scheme may adduct water from the neighboring grid cell with the largest streamflow volume. The choice of water adduction was introduced in Guimberteau et al. (2012b), but was disabled due to the coarse modelling resolution (grid-cell larger than 100x100 $km$ of size). Here we use a similar parameterization, but we add a user-defined parameter to take into account the facility to access distant



river reservoirs:

$$\frac{dS_{1,\text{add}}}{dt} = -Q_{1,\text{add}} = -\min\left(I_{\text{req}} - I, \frac{a_{1,\text{add}} S_{1,\text{add}}}{dt}\right) \tag{11}$$

In this equation, water adduction $Q_{1,\text{add}}$ from the largest river reservoir in the neighboring grid cell $S_{1,\text{add}}$, will depend on the facility of access represented by the factor $a_{1,\text{add}}$. This factor can range between 0 if there is no adduction, and 1 if the distant river reservoir is fully accessible for water adduction.

The irrigation water, $I + Q_{1,\text{add}}$, is finally added at the soil surface for infiltration, thus resembling a flood or drip irrigation technique. We note that irrigation is not restricted to an optimal period during the day, but may be triggered at any moment. It may lead to an overestimation of evapotranspiration (Ozdogan et al., 2010). We do not represent dams operation in this simulation, even if they play an important role to modulate the temporal dynamics of surface water and assure a water supply for irrigation in many large river basins (Pokhrel et al., 2016; Hanasaki et al., 2008a).

## 3    Data description

### 3.1    Input data for ORCHIDEE

Firstly, ORCHIDEE is run at global scale in offline mode. We run the model for the period 1970 - 2013, but we leave the first 10 years as warm-up, and we focus our analysis on the period 1980 - 2013. We use the GSWP3 (van den Hurk et al., 2016) as meteorological forcing (http://hydro.iis.u-tokyo.ac.jp/GSWP3/) with a resolution of 0.5 degrees. We also run short simulations
for the sensitivity analysis and parameter tuning (see below in Section 4). We prescribe the irrigated surfaces in transient mode, i.e. irrigated surfaces may change every year, based on the Historical Irrigation Dataset (HID) from Siebert et al. (2015) and on Land Use Harmonization 2 (LUHv2) dataset from Hurtt et al. (2020).

HID presents a map every 10 years before 1980 and every 5 years after at 5 arcmin resolution, and for each year, we use the nearest map in time to avoid data interpolation, LUHv2 presents a map every year with a 0.25 degrees resolution. The
main difference between the HID and LUHv2 maps is that HID prescribes the area that is equipped for irrigation (AEI), while LUHv2 prescribes the area that is actually irrigated (AAI). As a result, the HID dataset has a greater irrigated surface (3.0 $10^6 km^2$ for HID, 2.5 $10^6 km^2$ for LUHv2 at global scale around 2000).

The performed simulations use uniform parameters over irrigated areas, main changes between simulations are summarized in Table 1. As a reference, we use a simulation with no irrigation, called NoIrr, while simulation Irr with irrigation activated,
uses parameter values according to results from the sensitivity and tuning analysis (Section 4) and the HID maps. The simulation Irr_NoTuned also activates irrigation and uses the HID dataset as input, but it uses a-priori parameter values. This latter simulation does not consider the conclusions from the sensitivity analysis, and for instance does not activate irrigation withdrawal from the overland reservoir nor adduction.

We run additional simulations to assess the uncertainty of the simulated irrigation amount and the influence of the most sen-
sitive parameters, according to the sensitivity analysis: the impact of the desactivation of adduction in our scheme is considered in simulation Irr_NoAdd, the effects of changes in the $\beta$ value are considered in simulations Irr_NoTuned, Irr, and Irr_Beta,



**Table 1.** Simulations with inputs and parameter values. In brackets, the units of the parameter, [-] means that the parameter corresponds to a fraction and does not have a unit. In bold the change in parameter values respecto to the Irr simulation.

| Simulation | Irrigation | Irrigated surfaces | $\beta$ [-] | $A_i$ [-] | $I_{max}$ [mm/h] | Adduction [-] |
|---|---|---|---|---|---|---|
| NoIrr | **No** | – | – | – | – | – |
| Irr_NoTuned | Yes | HID | **1.0** | **0.9,0.0,0.9** | **1.0** | **0.0** |
| Irr | Yes | HID | 0.9 | 0.9,0.9,0.9 | 3.0 | 0.05 |
| Irr_LUH | Yes | **LUHv2** | 0.9 | 0.9,0.9,0.9 | 3.0 | 0.05 |
| Irr_NoAdd | Yes | HID | 0.9 | 0.9,0.9,0.9 | 3.0 | **0.0** |
| Irr_Beta | Yes | HID | **0.75** | 0.9,0.9,0.9 | 3.0 | 0.05 |
| Irr_Imax | Yes | HID | 0.9 | 0.9,0.9,0.9 | **1.0** | 0.05 |

and finally the effect of changes in the $I_{max}$ value is considered in simulation Irr_Imax. All these simulations use HID to prescribe irrigated areas. We analyze the effect of large differences in prescribed irrigated areas on irrigation amounts by using the LUHv2 dataset as input (simulation Irr_LUH) and using the same parameter values as the Irr simulation.

## 3.2 Validation datasets and landscape descriptors

The validation of the new irrigation scheme and its effect on the model bias is focused on five variables: evapotranspiration, leaf area index, discharge, irrigation withdrawal and total water storage anomalies. We also use two landscape descriptors datasets (see below).

**Irrigation water withdrawals -** We use two datasets: first, we compare the simulated irrigation rates with values from the
FAO-AQUASTAT database (https://www.fao.org/aquastat/en/) reported in Frenken and Gillet (2012) for irrigation volumes around 2000. AQUASTAT is based on reported values at the country scale, so it does not inform on seasonal values or their spatial distribution. In countries with a lack of information, data is completed using modelling outputs to estimate the plant requirement, and country-level ratios of irrigation efficiency to calculate the irrigation water withdrawal (Hoogeveen et al., 2015). While the plant requirement corresponds to the increase of evapotranspiration, the irrigation water withdrawal is the
volume that is abstracted from the natural reservoirs, and includes the losses and return flows.

We also use the spatially explicit information of irrigation water withdrawal around the year 2000 from Sacks et al. (2009). This reconstruction uses national-level census data, primarily from AQUASTAT, with maps of croplands by crop type, areas equipped for irrigation, and climatic water deficit. The result is a gridded map with a resolution of 0.5 degrees.

**Evapotranspiration -** We use two datasets: the first product is GLEAM v3.3a, which combines satellite-observed values
of soil moisture, vegetation optical depth, and snow-water equivalent, reanalysis of air temperature and radiation, and a multi-source precipitation product (Martens et al., 2017). The second dataset is FLUXCOM (Jung et al., 2019), which merges Fluxnet eddy covariance towers with remote sensing (RS) and meteorological (METEO) data using machine learning algorithms. Here



we use RS+METEO products, specifically the averages of RS+METEO<sub>WFDEI</sub> and RS+METEO<sub>CRUNCEP v8</sub>, to cover the analysis period.

**Leaf area index -** We use the LAI3g dataset (Zhu et al., 2013) climatological values for the period 1983-2015. This dataset applies a neural network algorithm on satellite observations of the Normalized Difference Vegetation Index (NDVI) 3g to estimate LAI.

**River discharge -** We use monthly data from the Global Runoff Data Centre (GRDC, https://www.bafg.de/GRDC-/EN/ Home/homepage_node.html) in 14 large basins with strong irrigation activities. We choose the station nearest to the river
mouth that also has data available for the study period (Figure S4 shows the basins and its corresponding discharge station). Basin boundaries were delineated with the flow directions map used by ORCHIDEE (section 2.1).

**Total water storage anomalies -** We compare the total water storage anomalies (TWSA) from our simulations with three different monthly products of TWSA based on GRACE (Gravity Recovery and Climate Experiments) observations based on global mascon solutions, that are suitable for hydrologic applications (Scanlon et al., 2016): CSR (Save et al., 2016), GRC
Tellus, called here TELLUS (Watkins et al., 2015) and NASA GSFC (Loomis et al., 2019). CSR has a spatial resolution of 0.25 degree, while TELLUS and NASA GSFC have a resolution of 0.5 degrees. As the differences between products at the large river basin scale are small, we use the average value of the three products. All the products cover the period from april 2002 to the end of the simulation in 2014.

**Landscape descriptors -** We compare the simulation results with two landscape descriptors which are linked to irrigation
and may contribute to the irrigation bias. We use the fraction of irrigated rice around year 2000 from MIRCA2000 (Portmann et al., 2010) (see spatial distribution and focus on Southeast Asia in Fig. S5), and the location and volume of major dams based on the Global Reservoir and Dams dataset, GRanD (Lehner et al., 2011). GRanD contains information on the maximum storage capacity and main use of dams with reservoirs larger than 0.1 ha. Here we consider dams that have irrigation as their main purpose.

## 3.3   Data processing and analysis

We aggregate and interpolate all the observed data to the 0.5 degrees spatial resolution of the ORCHIDEE simulations. For ET, we mask GLEAM and the simulated data according to FLUXCOM, which does not cover all the continents, so all the comparisons are made over the same grid cells with available information. For LAI, we exclude grid-cells with no data in LAI3g from the analysis. We compare grid cell values and zonal average values. The statistical significance of the mean
difference between observed and simulated time series is assessed with a Student's t-test at the 5 % significance level.

We use the simulated discharge from the grid cell that best matches the watershed area upstream of the discharge station. In addition, we only use time steps with data available from observations, so that both time series agree. For TWSA, we compare observed and simulated basin averages. As ORCHIDEE gives the total water storage (TWS) value, we normalize the time series with the mean value of the NoIrr simulation for the period 2002-2008, the same as the observed products. In this way,
the effect of irrigation over TWS is observed in the simulated time series.





In addition to direct comparison at the grid cell, zonal or basin scale, we perform a factor analysis to reveal relationships between modelling bias and landscape descriptors. We use the fraction of irrigated areas around 2000 from HID, as well as the fraction of irrigated rice from MIRCA2000, both interpolated to the ORCHIDEE resolution. We categorized grid cells into six classes by irrigation fraction levels based on the two datasets, following Mizuochi et al. (2021): Class 1: 0%, Class 2: 0 to 5%,

Class 3: 5 to 10%, Class 4: 10 to 20%, Class 5: 20 to 50% and Class 6: 50 to 100%.

We also performed a comparison between the average basin-scale irrigation bias and the volume capacity of dams used for irrigation within the basin, according to GRanD. We use the Pearson's correlation coefficient (r) as metric for the correlation analysis.

## 4   Sensitivity analysis and parameter tuning

### 4.1   Sensitivity analysis

Short simulations were run to assess the sensitivity of the irrigation amount at the global scale to different parameter values, assumed to be uniform in all irrigated areas. We used GSWP3 as meteorological forcing, and the LUHv2 from Hurtt et al. (2020) to prescribe the irrigated surfaces (see Section 3.1). We ran a total of 23 simulations with varying parameters, plus a reference simulation with no irrigation. All of them were run with the same initial conditions for three years (1998 - 2000),

and a comparison of irrigation amount and ET increase was performed for the year 2000. By using the last simulation year, we reduce the effect of the common initial conditions on the simulation results, and the year 2000 corresponds to the values given in AQUASTAT and Sacks et al. (2009). A brief description of each parameter as well as the unit, range, and values used in the sensitivity analysis is shown in Table 2.

We change the value of one parameter at a time (once-at-a-time screening, see Mishra (2009); Song et al. (2015)) and then

we observe its effect on the irrigation rate and on the increase in evapotranspiration. We tried to include the full range of parameters, but it is worth noting that in some cases, values were restricted to ensure an expected behavior. In the case of $\beta$, we set values around 1.0 (target equal to the field capacity soil moisture) as it seems a plausible target for flood irrigation, but note that values higher than 1.4 or lower than 0.6 are possible. Theoretically the upper limit is infinite, but values above 1.5 may exceed the saturated soil moisture for some soil textures, the lower limit is zero. For adduction, we set parameter values under

0.2 (20% of streamflow available for adduction at every time-step), which seem high enough to represent water adduction in large river basins (Leng et al., 2015). In the case of the $\text{LAI}_{\text{lim}}$ and $I_{max}$, upper values were selected a-priori. Those values in bold in Table 2 are called reference values afterwards. The reference parameter values are intended to maximize the irrigation amount, as preliminary tests (not shown) performed with a-priori values exhibited an underestimation of irrigation rates at global scale. The reference values do not change if not explicitly required by the once-at-a-time screening method.

Figure 2 shows that $\beta$ is the parameter with the strongest effect on the global mean irrigation rate, followed by the cumulated root density threshold $\text{Root}_{\text{lim}}$, $I_{max}$, and the fraction of stream storage available for adduction $a_{1,\text{add}}$. The fraction of water storage left for the ecosystems (called Environmental in the figure, $a_j$) has a more limited effect, suggesting that in many irrigated areas, there is enough water from surface and groundwater to fulfill the irrigation requirements. Finally, the LAI limit,





**Table 2.** Parameters of the irrigation module, brief description, range and values used in the sensitivity analysis. Values in bold correspond to the reference value. Note that, for parameter $A_i$, the three reservoirs share the same value.

| Parameter | Description | Unit | Range | Values |
|---|---|---|---|---|
| $\beta$ | Controls the soil moisture target, equal to $\theta_{fc} * \beta$ | Fraction, no units | $[0-\infty]$ | 0.6, 0.8, **0.9**, 1.0, 1.2, 1.4 |
| $a_j$ | Controls the fraction in reservoir available for irrigation, the complement being the fraction left for ecosystems | Fraction, no units | $[0-1]$ | 0.1, 0.5, **0.9**, 1.0 |
| $a_{1,add}$ | Controls the fraction in stream reservoir available for adduction | Fraction, no units | $[0-1]$ | 0.0, **0.05**, 0.1, 0.2 |
| $I_{max}$ | Maximum irrigation rate per hour | mm/h | $[0-\infty]$ | 0.5, 1.0, **3.0**, 5.0, 7.0 |
| $Root_{lim}$ | Defines if a soil layer is part of the root zone | Cumulated relative root density | $[0-1]$ | 0.0, 0.5 **0.9**, 1.0 |
| $LAI_{lim}$ | Minimum LAI in crops and grasses PFTs to trigger irrigation | m^2/m^2 | $[0-\infty]$ | 0.0, **0.1**, 0.3, 0.5, 1.0 |

$LAI_{\text{lim}}$, to trigger irrigation has a weaker effect than the other parameters. In the case of ET increase (Fig. 2, blue line), the
sensitivity to the different parameters is similarly hierarchized, although the magnitude is not necessarily the same. Also note that the effect on irrigation efficiency (i.e. ratio of ET increase and irrigation amount) is different for $\beta$ values higher than 1.0 and for $Root_{\text{lim}}$ values higher than 0.5. This implies that the fraction of irrigation water that becomes runoff or deep drainage is more important.

### 4.2 Parameter tuning

The sensitivity analysis showed that $\beta$ has the strongest effect on the simulated irrigation amount, and that these effects can induce changes on the irrigation efficiency. Therefore, we explored in more detail its behavior to set an value. Note that we used the chosen reference values for the other parameters. We compared the irrigation rate estimated by ORCHIDEE for 2000 in the short tests with the observed irrigation from Sacks et al. (2009) (Fig. 3) using total irrigation volume at global scale, and irrigation difference at grid-cell scale. When comparing the irrigation water amount at global scale (in $km^3$ for the year
2000, Fig. 3-a) we observe that a value of 1.2 maximizes the irrigation and minimizes the irrigation bias. When we assess the distribution of bias using grid-cell values (in mm/d, Fig. 3-b) we observe that for $\beta$ equal to 0.8, 0.9, or 1, the bias distribution is centered around 0, while it starts to move up for values 1.2 and 1.4. For simplicity here and as a tradeoff between the underestimation of irrigation volume and the spatial distribution of bias, we choose to set $\beta$ to 0.9 in all irrigated areas. We decided to use the reference values for the other parameters, as they play a minor role according to the sensitivity analysis, and
the reference values does not minimize the irrigation amount.



After this analysis we underline three points. First, this process does not correspond to a proper calibration, as we assumed uniform parameter values, the number of simulations is low and the observed data is sparse. The objective of the sensitivity analysis and parameter tuning was to identify key parameters and reduce the underestimation of irrigation by tuning the uniform parameter values. Second, although the once-at-a-time method is suitable given the computational cost of running 335 an ORCHIDEE simulation, it also has drawbacks and limitations in its analysis (Song et al., 2015), for instance its qualitative nature, and lack of quantification of individual interaction between parameters. Third, we use the LUHv2 map, which represents



**Figure 2.** Sensitivity of global irrigation volumes and increase of evapotranspiration ($km^3$) to changes in parameter values, for the year 2000 using short simulations. Secondary y-axis correspond to ET increase values compared to the simulation with no irrigation. Note that the y-axis scales differ between parameters.



the areas actually irrigated, AAI (a lower value than the areas equipped for irrigation, AEI, which is used in other datasets). We do not consider the effect of prescribed data uncertainty nor the effect of meteorological forcing in this analysis.

## 5  Results

### 340  5.1  Validation of irrigation water withdrawals

Irrigation from the Irr simulation is estimated at 0.049 mm/d (2452.5 km$^3$/year) around the year 2000 (Fig. 4-a). This estimation is in the lower part of other studies which range between 2465 and 3755 km$^3$/year (Pokhrel et al., 2016) and is lower than AQUASTAT estimation of 2735.1 km$^3$/year around the year 2000 (Frenken and Gillet, 2012). The results suggest that the proposed scheme is adequate to simulate the reported estimations of irrigation despite the underestimation (-10% than the 345 2735.1 km$^3$/year from AQUASTAT around 2000). We note that this estimate is also higher than that of the old irrigation scheme of Guimberteau et al. (2012b), but the scheme proposed here can still benefit from a more robust parameter tuning.

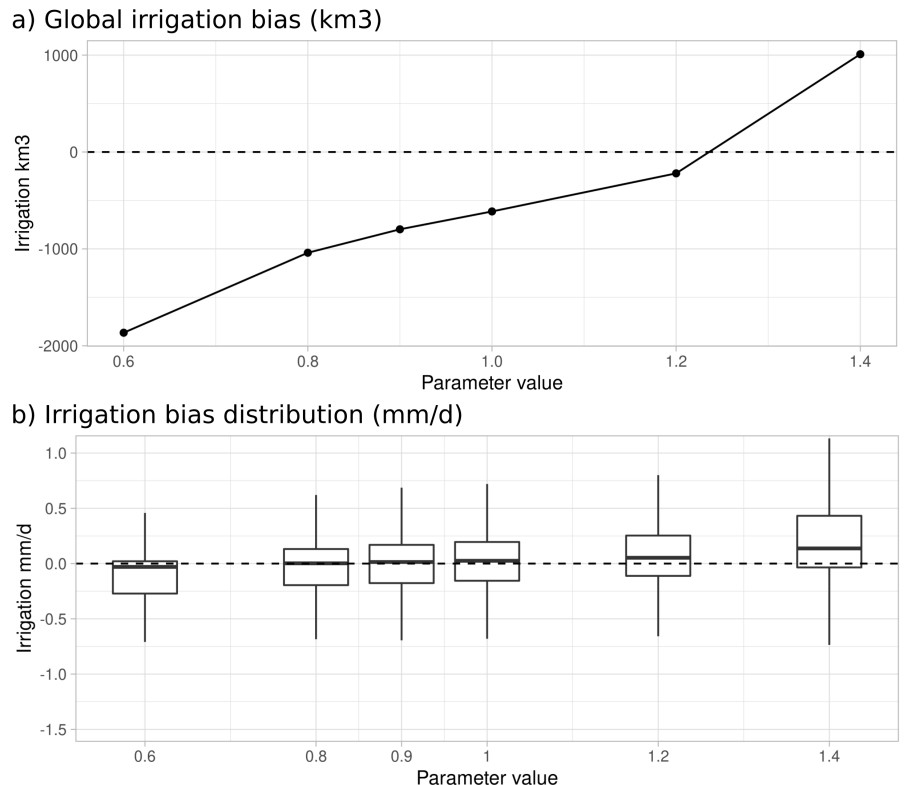

**Figure 3.** Calibration of $\beta$ value with Sacks et al. (2009) dataset as observed value, using outputs from the short simulations. Calibration of $\beta$ value using global irrigation volumes in $km^3$ (a) and boxplot using grid-cell bias in mm/d (b).



**Figure 4.** Total water withdrawal for irrigation in the Irr simulation, yearly average for 1998-2002 (a), groundwater irrigation withdrawn for irrigation, yearly average for 1998-2002 , difference in water withdrawn for irrigation between Irr (yearly average, 1998 - 2002) and AQUASTAT values (Frenken and Gillet, 2012) at country level in km3/year (b), difference in water withdrawn for irrigation between Irr (yearly average, 1998 - 2002) and dataset from Sacks et al. (2009) (c).

At the country-scale, Fig. 4-b shows that the irrigation module underestimates water withdrawals in the main hotspots of irrigation, i.e. India, China, and the USA, while it overestimate irrigation rates in Africa, East Europe and Latin America (see also Fig. S1). Such reduced contrasts between highly and weakly irrigated countries could indicate a limitation of the irrigation scheme to represent local irrigation strategies, as our scheme uses global uniform values for all the parameters. Comparison with the estimation from Sacks et al. (2009) (see Fig. S1) supports this result (-0.004 mm/d, -219.2 km3/y, Fig. 4-c) and allows us to identify the areas where the irrigation bias is the strongest. In India, the Indus basin presents a strong underestimation, as well as in the Northern part of the Ganges-Brahmaputra basin. In China, there is a more widespread underestimation. That is also the case in the west part of the US Great Plains. The other regions present in general an overestimation of irrigation





withdrawals, which is especially important in some small areas in Africa, in Eastern Europe and north to the Caspian sea, and in some areas of central Asia.

## 5.2 Variability of the irrigation rates due to parameter values and input data

The global annual irrigation volumes (Fig. 5-a) show a large uncertainty across the simulations due to changes in the parameter values (for instance, -24.7% between Irr_NoTunned and Irr). The parameter set used in the Irr simulation manages to increase

the irrigation rate and to markedly reduce the irrigation bias when compared to the Irr_NoTuned simulation. Also, a positive trend in the annual irrigation volume is observed in all simulations. It is caused by the increase in irrigated area, observed in both HID and LUHv2 datasets (see simulations Irr and Irr_LUH). The irrigated area has been identified by Puy et al. (2021) as the main driver of irrigation water withdrawal, and the increase of the prescribed irrigated area in the simulations explains in part the positive trend in the irrigation rate (see Fig. S6).

Based on the mean annual values (Fig. 5-c), the $\beta$ parameter has the largest effect on the mean irrigation rate (-22.3% when $\beta$ decreases from 0.9 to 0.75), followed by the change of input map from HID to LUHv2 (-19.7%), a lower $I_{max}$ (-16.5%) and, finally, no adduction (-15.7%). From a spatial point of view, the overall reduction in irrigation due to the above changes is not homogeneous and large areas may even display an increased irrigation rate. The exception is the $\beta$ parameter, which shows an overall reduction in irrigation with a lower parameter value, except in the Indus basin. The Indus river basin is a

region that depends on both surface and groundwater for irrigation, and the irrigation demand is one of the most important worldwide (Laghari et al., 2012). In Irr_Beta, a lower $\beta$ induces a reduction of water demand in the upper areas of the Indus river basin, increasing the river discharge downstream. More surface water supply in the middle and lower parts of the basin can increase irrigation in these areas, even if the water demand also decreases, because the irrigation deficit i.e. the difference between demand and supply, is still high despite the demand reduction. We advance here that the propagation of water supply

through the river system can explain part of the heterogeneity in the response to other parameter changes.

To estimate the interannual irrigation rate variablity, we calculate the coefficient of variation (ratio of standard deviation to the mean, Fig. 5-b). We use the pluriannual mean irrigation rate from all simulations with irrigation activated from table 1. Results highlight a certain homogeneity, but we can identify at least two distinct areas: an area of low variability (around 0.25) in Southern Asia, some areas in the Mediterranean and North and South America, and an area of high variability (around 0.75)

in Northern Europe, North America, Africa and Australia, with some points where the coefficient of variation is the highest (over 2.0), especially in Africa. The use of global parameter values could explain the relative homogeneity of the coefficient of variation, while regional differences like climate variablity and irrigation water demand could explain the existence of these two variability classes.

## 5.3 Factor analysis: correlation of modelling biases and irrigation classes

Figure 6-a shows the bias of ET by class of irrigated fraction at grid-cell scale, when we compare ORCHIDEE simulations with FLUXCOM dataset. It shows that the activation of irrigation reduces the ET bias in those areas with high irrigation fractions (also see S7 for the spatial distribution). For the comparison with GLEAM in Fig. 6-b, it shows that the activation of irrigation





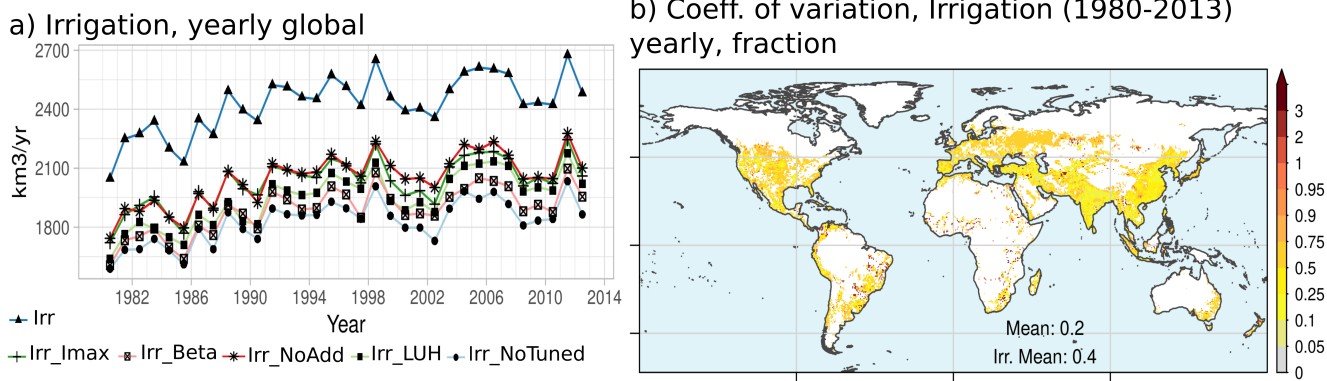

**Figure 5.** Time series of globally averaged irrigation rates simulated by ORCHIDEE (a). Map of the standard deviation of mean irrigation rates from all simulations in mm/d (b). Maps of mean difference between Irr simulation and others, for the period 1980 - 2013 in mm/d (c). Blank areas correspond to grid-cells with no irrigated areas





induces a positive bias in those areas with irrigation. When comparing absolute ET values by irrigation class (Fig. 6-c), we observe that NoIrr and GLEAM are similar in all the classes except for 0 and All, i.e. no irrigated fraction and all grid cells.

It means that the differences between NoIrr and GLEAM comes from non-irrigated areas. This could suggest a limitation in GLEAM to represent the effects of irrigation on ET rates as this product does not respond to the presence of irrigated areas. On the other hand, Irr and FLUXCOM boxplots are similar for classes 10-20, 20-50, and 50-100. A seasonal assessment on zonal average values for irrigated areas support this suggestion (Fig. S7). Thus, from now we prioritize FLUXCOM for our analysis regarding the ET bias.

A similar analysis for the LAI bias and classes of irrigated fraction (Fig 6-d) shows an increase in the LAI difference between ORCHIDEE and LAI3g in the Irr simulation. Also, for all classes, the positive bias in the NoIrr simulation is exacerbated in the Irr simulation, except for the most intensively class (class 50-100), which reduces the negative bias when comparing NoIrr and Irr simulations. It is worth noting that the class 50-100, where irrigation is more important, is the single one with a negative bias in NoIrr, and this negative bias is partially reduced when irrigation activities are included (see Fig. S8 for spatial distribution and

zonal average values). This is due to less water stress and thus more photosynthesis and biomass production, which is coherent with the decrease of ET bias for this class. When comparing absolute values between the simulations and the observed product (Fig. 6-e), we observe that irrigation activation within ORCHIDEE does not significantly change the distribution of LAI values at global scale. These results are coherent with changes that irrigation induces on water fluxes and reservoirs, as well as on water and energy budget (see Fig. S2 and S3).

## 405  5.4  Effect of irrigation on TWSA and river discharge

We now focus on the average TWSA value at the basin scale (Fig. 7). Activation of irrigation induces small changes in TWSA, which is coherent with changes in TWS between both simulations (Figure S2). For instance, we observe higher peaks and low values in Huang He when irrigation is activated, while in the Ganges river basin, low values are lower in the Irr simulation than in NoIrr. The changes in water pathways and related residence times that explains changes in TWS between Irr and NoIrr

(transfer of water from a reservoir with rapid flows like the streamflow to the soil, with a slow flow), also could explain these changes in TWSA dynamics at large basin scale. Other basins, like the Nile river basin or the Amu-Darya, show little effect between both simulations, even if extreme peaks values can be overestimated (during 2007 in the Nile) or underestimated (during 2005 and 2006 in Amu-Darya). But note that the model is unable to follow the GRACE trends, in basins with negative trends (for instance Huang He, Indus or Ganges) or positive trends (Murray river between 2011 and 2014).

The correct simulation of river discharge (Fig. 8) is another challenge in ORCHIDEE and other LSMs (Oki et al., 1999; Ducharne et al., 2003; Guimberteau et al., 2012a; Koirala et al., 2014; Cheruy et al., 2020). Irrigation plays an important role to reduce the average values when we compare NoIrr and Irr simulations (see for example the Nile, or the Indus rivers, these results are coherent with those from de Graaf et al. (2014)). The main effect of irrigation over the seasonal variations is that peak discharge can occurs before in the Irr simulation, or that the decrease after the peak is more rapid and low values are lower

in Irr than in NoIrr. These changes are due to the triggering of irrigation during spring and summer, and the corresponding ET increase. It is worth noting that the Irr simulation does not necessarily reduce the discharge bias against GRCD data compared





**Figure 6.** Factor analysis of ET bias with FLUXCOM against irrigated fraction classes (a), and with GLEAM (b). Mean ET values of simulations and observed products against irrigated fraction classes (c). LAI bais with LAI3g against irrigated fraction classes (d) and mean LAI values of simulations and observed product against irrigated fraction classes (e).

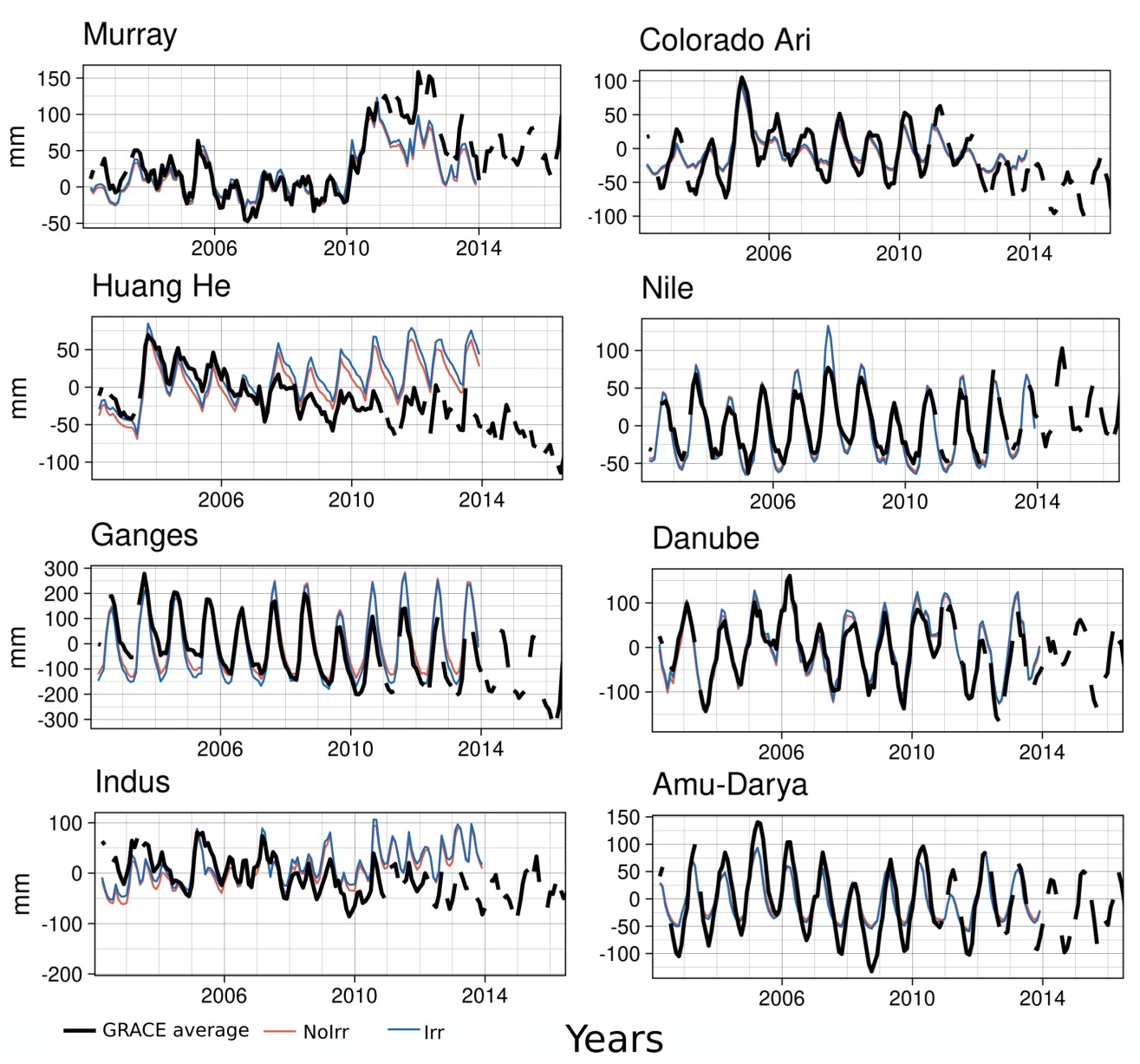

**Figure 7.** Comparison of TWSA between ORCHIDEE simulations and GRACE datasets in large basins with strong irrigation activities.





to the NoIrr simulation, with the exception of the Danube river (see Table S1 for some goodness-of-fit metrics for observed and simulated discharge values).

## 5.5 Factor analysis: correlation of irrigation biases and landscape descriptors

We compare biases and errors in irrigation estimates with landscape descriptors that could help explain these modelling errors. We also seek a perspective to increase the realism of the irrigation scheme and reduce the error in irrigation estimation. For the irrigation bias, classes with a high fraction of irrigated paddy rice, (for instance class 20-50 or 50-100), exhibit a higher bias than classes with small fractions (Fig 9-a). The spatial distribution of irrigated paddy rice is concentrated in southeast Asia, and includes the most irrigated river basins worldwide (see Fig. S5). At the large basin scale (see values in Table S2), the irrigation

bias also correlates well with the capacity of dams used for irrigation (Fig. 9-b) if we retire a single outlier corresponding to the Nile river basin (r value without the outlier is -0.55).

The correlation between paddy rice and irrigation bias suggests the need to explicitly represent paddy irrigation at global scale. Thus, we add an assessement of the $\beta$ value and irrigation bias, using the short simulations used on the parameter tuning (see section 4). We use all simulations with changes on the $\beta$ parameter from Table 2. Then, we build a composite map of the

$\beta$ value that minimizes the irrigation bias at a grid-cell scale (Fig. 9-c) and then we show the corresponding irrigation bias as compared to Sacks et al. (2009) dataset. The results roughly show at least two classes for $\beta$: the first with values of 1.2 and 1.4 (for instance in China and north India) and the second with values of 0.6. Using at least two $\beta$ values is not enough to reduce the irrigation bias at global scale, but it has an important effect on the spatial distribution of the irrigation bias in Southern Asia, the region with the most paddy rice area. These results suggest that the $\beta$ parameter should have at least two values, 1.3

in areas with paddy rice, and 0.6 in the rest of irrigated areas. But note that the data used for this analysis correspond to a single year, i.e. year 2000. Also, regional characteristics, like more than one harvest of paddy rice due to optimal climate conditions, are not taken into account in this analysis, but could also help to explain the irrigation underestimation in our estimations (Yin et al., 2020).

## 6 Discussion

In this study, we implemented a new global irrigation scheme inside the ORCHIDEE land surface model basedon previous work from Yin et al. (2020) in China. The model estimates the irrigation water demand by calculating a soil moisture deficit. Besides, it constrains the actual irrigation rate by the available water supply. The water supply takes into account the facility to access surface or underground water sources according to local infrastructure, and environmental restriction. Note that this environmental restriction is a simplification compared to the complex methods used in the real world to estimate environmental

flow requirements, and other more robust approaches exist (for instance in Hanasaki et al. (2008a), providing monthly environmental flow requirements). Strict environmental requirements could reduce the surface water supply, thus the irrigation rate (Hanasaki et al., 2008b).





**Figure 8.** Comparison of observed and simulated river discharge in large basins with strong irrigation activities.







**Figure 9.** Factor analysis of irrigation rate bias with data from Sacks et al. (2009), against irrigated paddy rice classes (a). Basin average value of irrigation bias against dams capacity (b). Map of $\beta$ that minimizes the irrigation bias, according to the short simulations (c) and corresponding map of minimum irrigation bias according to the $\beta$ value in (c), in mm/d (d).



For the facility to access the water sources, we use two static factors based on local infrastructure, while water allocation is dynamic and can change according to water availability (de Graaf et al., 2014) as well as economic and societal aspects

(D'Odorico et al., 2020). The irrigation scheme also allows to represent the adduction of water from neighboring grid cells, which can be important in areas of China and India (Laghari et al., 2012; Yin et al., 2021), where surface water is intensively used. This representation of water adduction, however, is very simple, and could be improved by including human water management and dams operation, as in Zhou et al. (2021), where the supply and demand network is operated as a system, taking into account some constrains like topography and environmental flow. Finally, the conditions to trigger irrigation, although

controlled by four parameters, may seem too simple in our scheme, especially compared to specialized irrigation models or the ISBA LSM Druel et al. (2022), which implement complex sets of rules to represent different irrigation strategies.

Despite this limitations, the evaluated irrigation scheme produces acceptable estimations of yearly irrigation withdrawals on a global mean basis, but it underestimates irrigation volumes in areas of China, India and the US (the most irrigated areas). Our estimations are affected by the uncertainty on global parameter values assumed to be uniform, and on the map of irrigated

fractions (Puy et al., 2021). We show that the lack of paddy rice irrigation could contribute to the underestimation of irrigation in southern Asia, as the paddy technique needs the inundation of the field and maintains a saturated soil at least during 80% of the crop duration (de Vrese and Hagemann, 2018). The irrigation module of LSM MATSIRO, called MAT-HI and HiWG-MAT (Pokhrel et al., 2012, 2015), already implemented an explicit representation of paddy rice irrigation, by setting a higher soil moisture target for rice than for other crops. An explicit paddy representation was also implemented in ORCHIDEE-CROP

(Yin et al., 2020) at a regional scale, by implementing pond for paddys and using a water level target, but it uses detailed crop information not easy to access at global scale. A surrogate approach in our simpler irrigation scheme could be to use at least two $\beta$ values, one for paddy rice and another one for other crops, as suggested by the composite map of $\beta$ values minimizing the irrigation bias.

An outcome of our study is to reveal that the GLEAM values do not exhibit a significant sensitivity of ET to the presence

of irrigated areas. This suggests that GLEAM is not suitable for estimating ET rates in irrigated areas. For instance, coupled simulations using CLM4 in northern India showed as strong modelling underestimation of ET rates, even with no irrigation (Fowler et al., 2018). When we compare the simulations with the FLUXCOM product, the activation of irrigation leads to a reduction of the negative evapotranspiration bias, but the use of a single soil column in ORCHIDEE for both rainfed and irrigated crops could induce an overestimation of ET increase. The ET bias improvement is particularly substantial in heavily

irrigated areas, where the simulated LAI is also improved by irrigation (which reduces there the negative LAI bias). These results show the benefits of including an irrigation scheme to partially reduce some modelling biases, especially in intensively irrigated areas, and are coherent with the multivariate evaluation of ORCHIDEE done in Mizuochi et al. (2021).

ET and LAI are two important drivers of land-atmosphere coupling via water, energy and momentum transfer (Seneviratne et al., 2010; Greve et al., 2019), but there is evidence that the effects on ET and LAI due to human land-cover change and

landscape management are not monotonic (Sterling et al., 2013). The sensitivity of these drivers to irrigation calls for further studies in coupled mode to explore the joint evolution of climate, land surface fluxes, and the use of water resources. Some studies focuses on the effects of irrigation on climate and land surface fluxes for the historical climate (Boucher et al., 2004;



Sacks et al., 2009; Puma and Cook, 2010; Guimberteau et al., 2012b; Cook et al., 2015; Thiery et al., 2017; Al-Yaari et al., 2022), but to the best of our knowledge, that is not the case for the future climate under different scenarios.

In contrast to the effects on ET and LAI, the effect of irrigation on land surface hydrology is rather weak. For discharge, activation of irrigation logically reduces river discharge, because of surface and groundwater withdrawal for irrigation. This reduction does not necessarily improves the model performance to fit observed values, with the exception of the Danube river basin. Multiple causes could explain the incorrect simulation of discharge dynamics in ORCHIDEE, even when irrigation is activated. For instance, uncertainties resulting from the atmospheric forcing are not assessed here, while they are known

to affect the yearly and seasonal values of discharge (Guimberteau et al., 2012a; Decharme et al., 2019). Also, a wrong ET estimation, errors in snow dynamics, and the lack of permafrost representation contribute to the mismatches (Cheruy et al., 2020). Finally, a lack of representation of other anthropogenic processes like dam management (Fig. 9), and water withdrawal for other economic sectors and other uses could explain the differences in seasonal discharge dynamics between ORCHIDEE and observed data in some basins (Pokhrel et al., 2016).

Effect of irrigation on simulated TWSA is weak. In some large river basins, we observed increases in low values in areas with significant surface water supply. But even when irrigation is activated, ORCHIDEE is not able to follow the trends exhibited by GRACE datasets, for instance in Huang He and Indus river basin, two heavily irrigated areas where water depletion has been related to groundwater pumping for irrigation (Rodell et al., 2018; Yin et al., 2020). There are probably multiple causes for inability of LSMs to capture large negative decadal water storage trends (Scanlon et al., 2018), starting with the underestimation

of irrigation rates at country-level and grid-cell scale (Fig. 4). Glacier loss misrepresentation in ORCHIDEE could also explain part of the differences to observed negative trends in some basins, for instance in the Indus and Ganges basins, that depend on water flow from the Himalaya mountains (Rodell et al., 2018). And of course, errors in the partitioning between the different water fluxes in ORCHIDEE (Cheruy et al., 2020; Mizuochi et al., 2021) contribute to the problems in both simulations (NoIrr and Irr).

We also underline the lack fossil groundwater abstraction in ORCHIDEE as a very likely cause to the underestimation of irrigation rates and TSWA trend mismatch. Fossil groundwater, also called non renewable groundwater, is important in semiarid areas like Pakistan and Middle East, and contributes nearly 20% to gross irrigation water demand for the year 2000 (Wada et al., 2012). As the irrigation scheme represents abstractions from shallow aquifers but not from fossil sources, it probably restrains irrigation too often due to a supply shortage, thus could have problems fitting the negative trend in those

areas with heavy groundwater use, as already reported by Yin et al. (2020) for China. But we must add that the estimation of fossil groundwater use is challenging. For instance, an assessment of the TWSA trends of residuals between our simulation and GRACE shows differences with estimates of groundwater depletion from (Wada et al., 2012) in some countries (see Table S3). Underestimation of irrigation rates, and uncertainties arising from fluxes partitioning and from meteorological data would also affect the estimations of fossil groundwater abstraction. So far, we cannot explain to which extent each one of these possible

causes participates to the misrepresentation of GRACE TWSA trends by ORCHIDEE.

Our results show that the new irrigation scheme helps simulating acceptable land surface conditions and fluxes in irrigated areas for ET and LAI, but they also show that inclusion of irrigation alone is not necessarily sufficient for a good fit between the





simulated values of TWSA and discharge and observed products. Including additional anthropogenic processes could help to reduce some of these biases. For instance, dams management and fossil irrigation withdrawal could increase the water supply
in some basins during dry months or years, thus increasing irrigation amount in areas with high irrigation demand and water supply shortage. At the same time, these processes may have an impact on river discharge dynamics and could help to represent the misrepresentation of TWSA trends in some areas.

## 7  Conclusions

We implemented a global irrigation scheme within ORCHIDEE LSM, with a simple representation of environmental restric-
tion, water allocation rules based on local infrastructure, and water adduction from non-local water reservoirs. We compared the irrigation estimates to reported values of irrigation withdrawal, and then we compared the outputs with and without irrigation to observed products of ET, LAI, TWSA and discharge. Our results highlight how the inclusion of irrigation can reduce some modelling biases, especially on ET and LAI, but they also underline the difficulties to represent irrigation on a large scale by using a simple scheme and limited information.

The model could still benefit from improvements on parameter tuning by explicitly representing paddy rice irrigation. Paddy irrigation could decrease irrigation bias in areas of southern Asia by increasing the irrigation demand. Dam management representation and inclusion of non-renewable groundwater use could also reduce negative biases in some heavily regulated basins by increasing the water supply. These three aspects could change the spatial distribution of the ET and LAI increases within the model. For TWSA and discharge, the inclusion of processes like dams management or fossil groundwater use could
help to represent observed seasonal dynamics and trends that the model is not currently able to represent.

   Finally, we remember that LSMs are commonly used in coupled mode with climate models and irrigation can have an impact on some atmospheric variables via changes on latent heat flux and leaf area index. Thus, the results obtained here encourage the use of coupled simulations to explore the joint evolution of climate under the ongoing climate change (for historical and especially for future periods), water resources, and irrigation activities. While there is an increasing litterature
body that explores the coupling of irrigation and climate for the historical period, to the best of our knowledge that is not the case for future scenarios. Coupled climate simulations for future scenarios coulp help to foresee potential changes on the joint long-term evolution of water resources use and climate, and might help to identify possible social consequences.

*Code availability.* The version of the ORCHIDEE LSM used for this study corresponds to tag 2.2, revision 7709 (Arboleda et al., 2023), and is freely available from https://forge.ipsl.jussieu.fr/orchidee/log/branches/ORCHIDEE_2_2/ (last access: 16 june 2023). It is provided under
a CECILL-C License (French equivalent to the L-GPL licence).

*Data availability.* The data from the ORCHIDEE simulations used for this study is freely accesible (Arboleda-Obando et al., 2023). Flux-com is available at https://www.bgc-jena.mpg.de/geodb/projects/Home.php after registration, GLEAM is available at https://www.gleam.eu/



after registration. LAI3g is available at https://daac.ornl.gov/cgi-bin/dsviewer.pl?ds_id=1653. Monthly discharge data from the GRDC is available at https://www.bafg.de/GRDC/EN/Home/homepage_node.html. For TWSA, CSR dataset is available at https://www2.csr.utexas.edu/grace/RL06_mascons.html, Tellus is available at https://podaac.jpl.nasa.gov/dataset/TELLUS_GRAC-GRFO_MASCON_CRI_GRID_RL06.1_V3, and GSFC is available at https://earth.gsfc.nasa.gov/geo/data/grace-mascons. We thanks William Sacks for providing the gridded irrigation rates dataset. HID dataset is available at https://mygeohub.org/publications/8/2 and LUHv2 is available at https://luh.umd.edu/. MIRCA2000 is available at https://zenodo.org/record/7422506. GRanD dataset is available at https://sedac.ciesin.columbia.edu/data/set/grand-v1-dams-rev01. For analysis, we used standard packages from R v.4.0.4 (R Core Team, 2016), https://www.R-project.org/

*Author contributions.* All the authors participated in the initial development of the irrigation scheme. PA and AD implemented the scheme in ORCHIDEE's code, run the simulations and evaluated the model. PA wrote the first draft. All the authors contributed to interpreting the results, discussing the findings and improving the final version of the paper.

*Competing interests.* The authors declare that they have no conflict of interest

*Acknowledgements.* This research was part of the PhD project of Pedro F. Arboleda, funded by EUR IPSL CGS (ANR Investissements d'avenir. Grant Number: ANR-11-IDEX-0004-17-EURE-0006). The work also received support from project BLUEGEM (ANR-21-SOIL-0001). The simulations were done using the IDRIS computational facilities (Institut du Développement et des Ressources en Informatique Scientifique, CNRS, France), under the allocation 2022-[AD010113599R1]. We thanks Bill Sacks for sharing the gridded maps on irrigation withdrawal for year 2000.



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
