# Peer review of "Validation of a new global irrigation scheme in the land surface model ORCHIDEE v2.2"

_EGUsphere, 2023_

## Author Comment (AC1)

**Response to Reviewers' observations on manuscript "Validation of a new global irrigation scheme in the land surface model ORCHIDEE v2.2"**

Pedro F. Arboleda-Obando, Agnès Ducharne, Zun Yin, Philippe Ciais
Correspondence: Pedro F. Arboleda-Obando (pedro.arboleda_obando@upmc.fr)

**Reply to anonymous reviewer 1**

Review comments on 'Validation of a new global irrigation scheme in the land surface model ORCHIDEE v2.2" by Arboleda-Obando et al.

The authors present a new global irrigation scheme inside the ORCHIDEE land surface model. The irrigation model calculates the irrigation water demand based on soil moisture deficit against their target soil moisture after irrigation, and the irrigation rate is constrained by the available water supply from three major reservoirs (stream, overland, and groundwater). Irrigation model parameter beta that controls the irrigation target soil moisture was tuned to match some existing global irrigation estimates. Global-scale irrigation estimate from the model is comparable with other existing estimates, e.g., FAO's AQUASTAT and Sacks et al. (2009), but its regional estimates show noticeable differences, particularly, underestimated irrigation in some irrigation hotspots in China, India and the US is notable. However, with the irrigation on, negative biases in ET over irrigated areas improved.

The new irrigation scheme shares common features with other some existing irrigation schemes that adopt similar concepts of adding irrigation water to soil up to a (tuneable) target value during the prescribed cropping seasons. But this work convincingly shows the importance of including irrigation scheme in global land surface (or hydrological) modelling to correctly reproduce evapotranspiration, which has important implications to relevant land surface processes and land-atmosphere interaction. Moreover, thanks to the explicit representations of irrigation water source, the authors argue the possible role of irrigation sourced from non-renewable groundwater storage in explaining the gap between modelled TWS and GRACE-derived TWS. The manuscript is well written and the topic is within the interests of EGUsphere's readers. I recommend that the manuscript is considered for publication in EGUsphere once some technical concerns listed in the following section are addressed.

We thank the anonymous reviewers for the time he spent to read and comment our paper. Below, we provide a point-by-point response to these comments. Sentences from the original submitted manuscript are presented in italic, while the proposition to respond to the observations are presented in bold. Observations from reviewer 1 are numbered from 1 to 20.

1. According to the description of irrigation scheme (Section 2.2), the soil moisture deific D is set to zero when crops and grasses are below a certain threshold value, LAI_lim. Although this might be a practical choice for the latter part of a crop growth cycle (maturity stage to harvest), irrigations from sowing the emergence stages, would be missed. Given that most crops require sufficient irrigation in the early stage of their growth cycle, this would lead to an underestimated irrigation overall.

We agree that the LAI threshold value could prevent irrigation during the early stages of the plant growth. As stated in the paper, we do so to prevent irrigation when there is no plant development, for example during boreal winter in the northern hemisphere. But we agree that it overlook irrigation during emergence, and could lead to underestimation. We propose to add the following sentence (in bold) to the paper, line 147 of the original submitted document:

*To prevent irrigation when there is no plant development, for example during winter, we set the deficit D to zero if all crops and grasses are below a certain LAI threshold, LAIlim.* **By doing so, we overlook irrigation used to enhance germination, and tend to underestimate irrigation amounts.**

Besides, we add a sentence in the discussion, line 459, to take into account this observation and observation 3 from reviewer 1 (see response to observation 3 from reviewer 1)

2. In addition, it is stated in Section 2.2 that "*we do not separate the irrigated area into a separate soil column, i.e. the soil column includes crops (both irrigated and rainfed) and grasses…The effective irrigation (I, see below) is uniformly applied over the crops and grasses soil column.*" This implies that if a fraction (<1) of crop/grass column is irrigated to meet the target soil moisture content (beta x field capacity), the added water is spread over the whole column, leading to the soil moisture content still under the target soil moisture. This will in turn make the model add more water for irrigation until the entire column that contains fractional crops/grasses receives water up to the target soil moisture. Is this the case? The authors mention overestimated evapotranspiration as a possible result of the simplified water addition scheme, but in combination with the additional irrigation caused by the uniform spread of water to the entire crop/grass column, over-irrigation effect can be fairly significant, particularly when a crop/grass column represent sparse cropping area with a small crop fraction.

The reviewer is right. In the case that the irrigated area is much smaller than the crops and grasses soil column fraction, the added water does not necessarily lead to increase the soil moisture content over the target soil moisture. The result is continuous irrigation (if there is soil deficit and available water) during the growing season. But note that the irrigated fraction and the maximum irrigation rate per hour also control the water demand. So even if overestimation is a likely output, underestimation is also possible.

We propose to change the sentence in line 159 of the original submitted document, so this particular case is its consequences are better explained:

*If the fraction of irrigated area is **much** smaller than the fraction of the crops and grasses fraction of soil column, irrigation will eventually be **spread** over a larger area than the actually irrigated surface. **This particular case (an important difference in irrigated fraction and soil column fraction) could likely result in overestimation of the amount of irrigation (mainly because the water put on the surface will not be sufficient to reach the soil moisture target, see S9). Besides, the** fraction of irrigation water **that** actually **evaporates could be larger** than in reality. **The latter** could lead to an overestimation of the evapotranspiration increase, especially in areas that are energy-controlled \citep{Puma2010} **and an overestimation of irrigation efficiency.***

Also, we propose to add a new figure in supplementary, Fig. S9. Fig. S9-b shows the irrigation bias by class of irrigated surface over crop and grass soil column. In comparison, in Fig. S9-a we show the same plot by class of irrigated fraction. While sparse irrigation in croplands could likely lead to overestimation in our model, it is not always the case. Also, underestimation in intensively irrigated areas is more important.

The new Figure S9 is shown below:

[Figure]

*Figure S9 Use of factor analysis against irrigation bias. Irrigation rate bias against data from Sacks et al. (2009), as a function of irrigated fraction classes (a) and classes of the ratio of irrigated fraction and crops and grasses soil column fraction (b). Both plots use data from the Irr simulation for 2000. Irrigation rate bias against data from Sacks et al. (2009), as a function of irrigated fraction classes and 'beta' parameter values (c). Irrigation rate as a function of irrigated fraction classes and 'beta' parameter values (d). Both plots (c) and (d) use data from short simulations used for the tuning parameter analysis, for 2000.*

3. The new irrigation scheme add irrigation water to close to soil moisture deficit, the difference between the actual soil moisture and 'beta x field capacity', but the manuscript does not provide the soil moisture value that triggers irrigation (this is a different trigger than the LAI_min). Does this mean that irrigation is triggered whenever soil moisture drops below 'beta x field capacity'? This would result in continuous irrigation over the whole duration of the cropping season (when LAI >

LAI_min), leading to unrealistic emulation of irrigation (and likely overestimation of irrigation).

The reviewer is right, the use of a 'beta x field capacity SM' might result in a continuous irrigation during the whole growing season (if there is enough available water), unless soil moisture becomes greater than the target due to precipitation.

To address this observation, we propose to include all the limitations of the scheme at the beginning of discussion (see response to observation 13 of the first reviewer).

We propose to add some clarifications in line 147 of the original submitted document:

*where Wi and Wfc i(both in mm) are the actual and field capacity soil moisture in soil layer i, respectively, and β is a user dependent parameter that controls the target value **with respect to field capacity (see Fig. S10 in the supplementary for information regarding soil texture and field capacity soil moisture in the root zone). When soil moisture drops below the target, irrigation is triggered.***

Besides, in order to take into account observations 1 and 3 from reviewer 1, we add a sentence in the discussion, line 459 of the original submitted document:

*Finally, the conditions to trigger irrigation, although controlled by four parameters, may seem too simple in our scheme, especially compared to specialized irrigation models, **the new irrigation scheme in LSM CLM5 (Yao et al., 2022), or the ISBA LSM (Druel et al., 2022),** which implement complex sets of rules to represent different irrigation strategies. **Some rules could change the moment when irrigation is triggered and increase the amount (for instance allowing irrigation some days before the crop emergence) or decrease the irrigation amount (for instance, preventing irrigation during maturity of the crop, or preventing continuous irrigation during more than a certain number of days).***

Finally, we propose to add a map of soil texture as used by the model, and the corresponding field capacity SM, in Fig. S10 of the supplementary, and cite it in line 147.

[Figure]

**Figure S10 Soil texture map used by ORCHIDEE from Zobler (a) and field capacity soil moisture in the root zone defined according to the new irrigation scheme, in kg/m2 or mm. Both maps are at the simulation resolution (0.5° x 0.5°). White in (b) means that there is no irrigated fraction according to HID map for year 2000.**

Specific Comments

4. Line 43: "…potential evapotranspiration PET, ET0…" -> (PET)?

The ETc corresponds to the crop-specific potential evapotranspiration, and is equal to ET0 = kc · PET, where kc is a crop-type and growing stages dependent parameter, and PET is the atmospheric evaporative demand. We propose to slightly change this paragraph so it is more comprehensive:

*This ET increase is estimated as the differences between crop-specific potential **ET** and actual ET with no irrigation (Siebert and Döll, 2010; Mekonnen and Hoekstra, 2011; Wada and Bierkens, 2014; Chiarelli et al., 2020).* ***Following Allen et al., 1998, the crop specific potential ET is defined as ETc = kc · ET₀,*** *where **parameter** kc **depends on** crop-type and growing stage, **and ET₀ is the reference crop ET, corresponding to the atmospheric evaporative demand**.*

5. Line 44-45: "This reference PET...parameter." This sentence needs to be rewritten for clarity.

We propose to change this phrase, see the response 4. below.

6. Line 125: "Tree" -> Three

We propose to correct the typo in the new version of the paper.

7. Line 138: Correct "crop- grass soil column"

We propose to change the typo in the new version of the paper.

8. Line 156-158: For the reason described in the general comment section, I think 'the fraction irrigated is greater than the crop/grass soil column' would be less of a concern for correction irrigation simulation.

We agree, but it belongs to the checks currently made by the scheme, so we prefer to leave it in the manuscript.

9. Line 166: What is the definition of 'renewable-groundwater reservoirs' in this work?

It corresponds to shallow aquifers recharged by drainage at the soil bottom. We propose to add a sentence in the discussion, line 166 of the original submitted document:

*In this equation, Sj [mm] is the volume storage in each routing reservoir, with index j equal 1, 2 and 3 for the stream, overland, and renewable-groundwater **(i.e. shallow aquifers that are recharged by drainage at the soil bottom)** reservoirs, respectively.*

10. Line 193: "100x 100 km" -> "100 x 100 km"

We propose to change the typo in the new version of the paper

11. Line 214-217: 3.0 x 10^6, 2.5 x 10^6. Does HID include LUHv2 or separate? It appears to be assumed in two different ways?

If both datasets shared the same spatial distribution, AEI should include AAI and so, HID should include LUHv2. But as both datasets rely on different information sources, processed with different methods, inclusion of AAI in AAI is not the case, in fact it is easy to verify the important differences in the spatial distribution of both datasets.

We include a sentence to clarify these differences between both datasets:

*The main difference between the HID and LUHv2 maps is that HID prescribes the area that is equipped for irrigation (AEI), while LUHv2 prescribes the area that is actually irrigated (AAI). As a result, the HID dataset has a greater irrigated surface (3.0 $10^6 km^2$ for HID, 2.5 $10^6 km^2$ for LUHv2 at global scale around 2000). It also means that AAI should be included in the AEI if the two datasets shared a similar spatial distribution. **But this is not the case, as the two datasets rely on different information sources, processed with different methods (Oliveira 2022).***

12. Line 221: "a-priori" -> *a priroi*

We propose to change the typo (from a-priori to *a priori*) in the new version of the paper

13. Line 320-321: This justifies a close exam of threshold SM triggering irrigation and possible flaws in irrigation application to the entire crop/grass column with a fractional coverage, particularly when the fraction is small.

It is related to the effect of beta on water amount and irrigation efficiency

It could be related to some flaws on the irrigation scheme, but also to the representation of infiltration, or to other model flaws, like an overestimation of bare soil evaporation.

We agree with the reviewer, it is possible that part of the irrigation bias is probably due to flaws in the scheme, to the effect of the tuned 'beta' value, and to other flaws in the model, for instance infiltration.

We propose to include a classification of the type of limitations that could impact our estimates of irrigation amount or the effect of irrigation on other variables, at the beginning of the discussion, line 445 of the original manuscript:

*In this study, we implemented a new global irrigation scheme inside the ORCHIDEE land surface model based on previous work from Yin et al. (2020) in China. **While we found a reduction in some modelling biases when irrigation is activated, we also identified at least four types of limitations in our modelling framework that can affect the estimates of irrigation or the effects of irrigation on other variables inside the land surface model:***

1. ***The irrigation scheme exhibits some shortcomings that may bias the estimated irrigation amount: the use of a single irrigation technique; simplified rules to trigger irrigation and allocate the available water; the joint representation of rainfed and irrigated crops within the same soil column; the non-representation of conveyance losses, although losses due to return flows are represented.***
2. ***The parameter tuning is overly simplistic. As a first step, we considered globally uniform parameters, which is overly simplistic, although spatially distributed values would allow us to better describe the local features of irrigation systems, as shown by the spatial variations in optimized β map, and the dependence of the local irrigation bias on the fraction of paddy rice.***
3. ***We also use a single meteorological forcing dataset and a single year to characterize observed irrigation values. This contributes to biasing the parameter adjustment process by taking uncertain data (meteorological forcing and reference irrigation) as certain.***
4. ***The ORCHIDEE model exhibits many uncertainties that are not related to the irrigation scheme, but ultimately impact the irrigation withdrawals and efficiency (defined here as the ratio of additional ET due to irrigation to water withdrawal) and the temporal dynamics of irrigation. One particular uncertainty comes from the overestimation of bare soil evaporation (Cheruy et al., 2022),***

*that we presently try to correct in ORCHIDEE. Other uncertainties result from the inherent simplifications of any model. In ORCHIDEE, they include the use of a single soil texture in each grid cell, of only two kinds of crops with simplified phenology and crop calendars, and the choices made to simulate infiltration and evaporative processes.*

Finally, we propose to add two additional plots to Fig. S9. Fig. S9-c shows the effect of changing beta on irrigation bias, by class of irrigated fraction. Fig. S9-d shows the effect of changing beta on the irrigation rate, by class of irrigated fraction (see Fig. S9 in response to observation 2 reviewer 1). Both show that higher beta increases the irrigation rate, but the effect on irrigation bias is different according to the class.

We propose to include the next sentence in bold in line 328 of the original manuscript to cite the new Figure:

*When comparing the irrigation water amount at global scale (in km3 for the year 325 2000, Fig. 3-a) we observe that a value of 1.2 maximizes the irrigation and minimizes the irrigation bias. When we assess the distribution of bias using grid-cell values (in mm/d, Fig. 3-b) we observe that for β equal to 0.8, 0.9, or 1, the bias distribution is centered around 0, while it starts to move up for values 1.2 and 1.4.* **This behavior can be slightly different depending on the irrigated fraction (see Fig S9)**.

14. Line 341-342: Irr simulation result, 2452.5 km^3/y appears to be closer to the higher end of 3755-2465? Also results in Figure 4 indicate that the difference in global irrigation may not reflect that large continental/country scale difference.

We do not fully understand, but our result for total irrigation withdrawal, 2452, is in the low part of the range 3755-2465 $km^3$/y. We agree that the differences in global irrigation (around -10% compared to the irrigation amount of ~2700 $km^3$/y from AQUASTAT) does not necessarily reflect large differences at continental/country scale because the over and underestimations largely balance each other. That is why we include an evaluation at country level, and at grid-cell level.

To make this point clearer, especially for the differences within a country, we propose to add a sentence in the discussion, line 347 of the original submitted document, and cite Fig S11 (See response to observation 7 reviewer 2 for the Figure):

*The other regions present in general an overestimation of irrigation withdrawals, which is especially important in some small areas in Africa, in Eastern Europe and north to the Caspian sea, and in some areas of central Asia.* **Finally, we note that within a country, it is possible to observe areas with positive and negative bias, for instance in the USA or India. This could also be partially explained by the use of globally uniform values, as there could be important local differences on irrigation strategies within the same country, and it remarks the need to assess the irrigation bias at different scales (See Fig. S11).**

15. Figure 3 caption: The caption describes what authors did with the beta vs. irr, but it does not properly describe what the figures are about.

To clarify the figure's description, we propose to change the caption to the following one:

*Figure 3. Calibration of β value with Sacks et al. (2009) dataset as observed value, using outputs from the short simulations.* **Bias in total irrigation volume in km³ by β value (a), boxplot of the bias of irrigation rates in gridcells in mm/d by β value (b).**

16. Line 349-350: This is not a convincing explanation because Figure 4c shows noticeable contrasts between over- and under-irrigation within a country, for example in the US and India.

We agree with the reviewer, and we included a clarification in that sense (see response to observation 14 reviewer 1)

17. Line 358-359: Again, global annual irrigation may not be the best way to quantify errors in irr when they show strong biases with different signs between continents/countries.

We agree with the reviewer, as said in response 14 and 16, and that is why we include an evaluation at different scales.

We propose to add the following two sentence after line 359 of the original submitted paper so the spatial heterogeneity of the variability is highlighted in this part of the manuscript:

*The global annual irrigation volumes (Fig. 5-a) show a large uncertainty across the simulations due to changes in the parameter values (for instance, -24.7% between Irr_NoTunned and Irr)*, **but note that the change on irrigation rates at gridcell scale can have a strong spatial heterogeneity within a country (Fig. 5-c) for instance in India or the USA.** *The parameter set used in the Irr simulation manages to increase the irrigation rate and to markedly reduce the irrigation bias when compared to the Irr_NoTuned simulation* **at global scale, even if locally we may observe both an increase or a decrease on the irrigation rate in the same country, for instance in China (with a marked north-south difference, Fig. 5-c, Irr_NoTuned-Irr) or the Indus river basin in Pakistan and India (see Fig. 5-c, Irr_NoTuned-Irr).**

18. Line 436-437: I am not sure what the model would do with a beta > 1. Since soil moisture would be max at the (effective) porosity, if beta x field capacity > porosity, would the irrigation scheme keep adding water every time step?

The reviewer is right, if beta x field capacity > porosity, the scheme would keep adding water every time step (if water is available), but we care that beta x field capacity < porosity by choosing the beta value. We propose to show the theoretical maximum 'beta' by texture in the supplementary. The maximum 'beta' results of the ratio of saturated soil moisture to field capacity soil moisture, as shown in table S4.

We cite the new table in line 304 of the submitted manuscript

*Theoretically the upper limit is infinite, but values above 1.5 may exceed the saturated soil moisture for some soil textures, the lower limit is zero* **(see Table S4).**

**Table S4 Parameters for soil textures used in ORCHIDEE. θs is saturated soil moisture and θfc is the field capacity soil moisture, in volumetric content. β max is the theoretical maximum β value that can be used, where β = θs/θfc , so the target soil moisture does not surpass saturated conditions.**

| Parameter | Sandy Loam | Loam | Clay Loam |
|-----------|-----------|--------|-----------|
| θs (m3/m3) | 0.41 | 0.43 | 0.41 |
| θfc (m3/m3) | 0.1218 | 0.1654 | 0.2697 |
| β max (-) | 3.4 | 2.6 | 1.5 |

19. Line 445: "basedon" -> based on

We will correct this typo in the new version of the manuscript.

20. Figure 9 inset text: Reduce the font size to make the whole text visible.

We corrected a small typo in the inset text, but we do not fully understand which text is not visible in this figure. We propose to discuss with the editor to correct this observation.

**Final note:**

We also corrected the style of some sentences, so they were more understandable, and some orthographic and grammar errors.

**REFERENCES**

Allen, R. G., Pereira, L. S., Raes, D., and Smith, M.: Crop evapotranspiration, Irrig. Drainage Paper, 56, UN Food and Agriculture Organization, Rome, Italy, ISBN 92-5-104219-5, 1998.

Cheruy, F., Ducharne, A., Hourdin, F., Musat, I., Vignon, E., Gastineau, G., Bastrikov, V., Vuichard, N., Diallo, B., Dufresne, J., Ghattas, J., Grandpeix, J., Idelkadi, A., Mellul, L., Maignan, F., Menegoz, M., Ottlé, C., Peylin, P., Servonnat, J., Wang, F., and Zhao, Y.: Improved near surface continental climate in IPSL-CM6A-LR by combined evolutions of atmospheric and land surface physics, Journal of Advances in Modeling Earth Systems, https://doi.org/10.1029/2019MS002005, 2020.

Chiarelli, D. D., Passera, C., Rosa, L., Davis, K. F., D'Odorico, P., and Rulli, M. C.: The green and blue crop water requirement WATNEEDS model and its global gridded outputs, Scientific Data, 7, 1–9, https://doi.org/10.1038/s41597-020-00612-0, 2020.

Druel, A., Munier, S., Mucia, A., Alberger, C., Calvet, J. C.: Implementation of a new crop phenology and irrigation scheme in the ISBA land surface model using SURFEX_v8.1, Geosci. Model Dev., 15, 8453–8471, 10.5194/gmd-15-8453-2022, 2022.

Mekonnen, M. M. and Hoekstra, A. Y.: The green, blue and grey water footprint of crops and derived crop products, Hydrology and Earth System Sciences, 15, 1577–1600, https://doi.org/10.5194/hess-15-1577-2011, 2011.

Sacks, W. J., Cook, B. I., Buenning, N., Levis, S., and Helkowski, J. H.: Effects of global irrigation on the near-surface climate, Climate Dynamics, 33, 159–175, https://doi.org/10.1007/s00382-008-0445-z, 2009.

Siebert, S. and Döll, P.: Quantifying blue and green virtual water contents in global crop production as well as potential production losses without irrigation, Journal of Hydrology, 384, 198–217, https://doi.org/10.1016/j.jhydrol.2009.07.031, 2010.

Wada, Y. and Bierkens, M. F. P.: Sustainability of global water use: past reconstruction and future projections, Environmental Research Letters, 9, 104 003, https://doi.org/10.1088/1748-9326/9/10/104003, 2014.

Yao, Y., Vanderkelen, I., Lombardozzi, D., Swenson, S., Lawrence, D., Jägermeyr, J., et al.: Implementation and evaluation of irrigation techniques in the Community Land Model, Journal of Advances in Modeling Earth Systems, 14, https://doi.org/10.1029/2022MS003074, 2022.

Yin, Z., Wang, X. H., Ottlé, C., Zhou, F., Guimberteau, M., Polcher, J., Peng, S. S., Piao, S. L., Li, L., Bo, Y., Chen, X. L., Zhou, X. D., Kim, H., and Ciais, P.: Improvement of the Irrigation Scheme in the ORCHIDEE Land Surface Model and Impacts of Irrigation on Regional Water Budgets Over China, Journal of Advances in Modeling Earth Systems, 12, 1–20, https://doi.org/10.1029/2019MS001770, 2020.

---

## Author Comment (AC2)

**Response to Reviewers' observations on manuscript "Validation of a new global irrigation scheme in the land surface model ORCHIDEE v2.2"**

Pedro F. Arboleda-Obando, Agnès Ducharne, Zun Yin, Philippe Ciais
Correspondence: Pedro F. Arboleda-Obando (pedro.arboleda_obando@upmc.fr)

**Reply to anonymous reviewer 2**

Review of "Validation of a new global irrigation scheme in the land surface model ORCHIDEE v2.2" by Arboleda-Obando et al. for GMD

The authors improved an irrigation scheme in LSM ORCHIDEE and evaluated the improvement. Using reported irrigation statistics for the year 2000, they globally-uniformly tuned key parameters of the irrigation scheme used in ORCHIDEE to achieve a better balance in estimating the global total irrigation volume and spatially minimizing irrigation bias. It is also investigated how each of these parameters can change the irrigation estimate. In addition to modifying their irrigation scheme, this study shows how much irrigation can affect simulations on hydrological processes in terms of several hydrological variables: evapotranspiration, leaf area index, river discharge and total water storage. In addition, their factor analyses indicate potential research directions to further improve the ORCHIDEE irrigation model, such as the explicit inclusion of paddy rice.

Such model improvement is essential for a better understanding of land surface processes in Earth system science. Considering the fact that human activities have influenced the Earth system, irrigation should also be a critical component to be further investigated. I understand that this is an important step for ORCHIDEE. However, I have some major concerns that the authors need to address.

We thank the reviewer for his time on reading and reviewing this manuscript. Below, we provide a point-by-point response to these comments. Sentences from the original submitted manuscript are presented in italic, while the proposition to respond to the observations are presented in bold. Observations from reviewer 2 are numbered from 1 to 19.

< Major comments >

1. (1) Why do the authors insist on globally uniform parameters (tuning)? While better estimation of total global irrigation withdrawals is an important challenge, better estimation of irrigation in heavily irrigated regions should also be a priority in a global study. The results show that this irrigation water estimate is relatively small compared to other irrigation estimates (Section5.1), and this should be related to the underestimation of irrigation volume in heavily irrigated countries (Fig4b-c). On the other hand, the globally uniform parameter tuning reduced irrigation volume to exacerbate the underestimation in these regions (Fig5c-1, Fig9). The authors also state that this is a drawback (Line.331). Therefore, I wonder if this tuning is sufficient to improve the simulation skill of ORCHIDEE. Since the authors already present

spatially varying beta value, which is a key tuned parameter, to minimize the irrigation bias in this study (Fig9), I wonder why the authors did not apply this spatially varying parameter to estimate the main irrigation estimate. I assume that there are reasons (perhaps, related to modeling philosophy) for this decision to apply globally uniform parameters and their tuning. If so, I expect the authors to clarify their thoughts in an earlier part of this manuscript.

We agree with the reviewer, better estimations of irrigation withdrawals at global scale are not incompatible with better estimations of withdrawals in those heavily irrigated regions. But we insist on globally uniform parameters, because: it is a first step in the implementation of irrigation at global scale; it is a tradeoff between representation of first-order effects and inclusion of local irrigation strategies; despite the shortcoming, we show that the scheme adequately represent main effects; finally we show that there is an interesting clue to improve irrigation representation by including a spatially varying 'beta' value, using the presence of paddy rice fields.

The latter (using paddy rice to spatially vary 'beta') is important since punctual optimization is not considered in ORCHIDEE due to scale dependence, so the LSM needs to use observed spatial features (e.g., fractions of PFT, soil texture, or paddy rice fractions).

We propose to include the drawback of globally uniform parameters and the limitation on the parameter tuning at the beginning of the discussion (section 6 of the original manuscript), as described in the answer to observation 13, reviewer 1.

2. (2) Another concern related to the parameter tuning is the reference year, 2000. Given the spatiotemporal uncertainty in the meteorological forcing data (even with reanalysis-based forcing data), I wonder if the single reference year allows the authors to robustly tune the parameter. I require an explanation or methodological modification in this regard.

These two concerns (uncertainty in meteorological data, and uncertainty in the tuning process with a single reference year) are important. We agree with the reviewer that these two issues could affect the robustness of the parameter tuning.

On the first issue, it was out of the paper scope to assess the effect of changing the meteorological input on irrigation estimates, although we expect to include this effect in the future. On the second issue, the year 2000 for tuning is often used as reference, because there is more data for that year (see for instance Pokhrel et al., 2016, table 2)

We propose to include these two issues at the beginning of the discussion (section 6 of the original manuscript), as **described in the answer to observation 13, reviewer 1**.

We propose to add a sentence on the concern about the use of a single reference year and the reasons to use data from year 2000 (l. 295 of the original manuscript).

*We ran a total of 23 simulations with varying parameters, plus a reference simulation with no irrigation. All of them were run with the same initial conditions for three years (1998 - 2000), and a comparison of irrigation amount and ET increase was performed for the year 2000. By using the last simulation year, we reduce the effect of the common initial conditions on the*

*simulation results, and the year 2000 corresponds to the values given in AQUASTAT and Sacks et al. (2009).* **Note that we use a single meteorological forcing dataset and compare our estimates to a single set of observed AQUASTAT data for the period around the year 2000. We choose to compare our estimates for the year 2000 because this year is commonly used as the reference period in the literature concerning the estimation of the amount of irrigation on a global scale (see, e.g., Pokhrel et al., 2016, Table 2). The choice of the year 2000 is mainly due to the existence of more complete reported or observed values for that year, as well as simulated estimates. We use the same reference period to compare our results with independent data.** *A brief description of each parameter as well as the unit, range, and values used in the sensitivity analysis is shown in Table 2.*

3. (3) I would like the authors to revisit irrigation efficiency and describe in more detail how they account for this factor in their irrigation estimate. I may be wrong, but as far as I have read Section 2.2, evaporative, infiltration, and seepage losses during conveyance, distribution, and application processes, are not considered in the calculation of irrigation water withdrawal from irrigation requirement. Other models that use soil moisture target methods generally consider irrigation efficiency (such as CLM5, LPJmL, H08, and HiGWMAT etc.). Although irrigation efficiency has a large uncertainty, if irrigation efficiency is not considered, irrigation withdrawal cannot be properly estimated from irrigation requirements. Thus, the relatively smaller global total irrigation volume estimated in this study may be related to this point. (Note that this is the different irrigation efficiency defined in line316.)

The scheme does not consider conveyance losses,  but we do allow the model to calculate losses related to the application of irrigation, as we simply put the withdrawn water over the soil surface for infiltration. This means that the model decides if the added water runs off or not, then if the infiltrated water is used by the plant or evaporates from the bare soil, or if ultimately it increases deep drainage. It also implies that the water demand is rarely fulfilled during a single time step, because not all the added water is used by the plant. That is why we fit our values to reported irrigation amounts from AQUASTAT, but check the effect in evapotranspiration increase as well.

We propose to show how irrigation efficiency is considered in some models, in introduction, line 45 of the original manuscript:

***Some models*** *also consider* ***conveyance*** *losses and* ***return flows to rivers and aquifers****, i.e. they consider the total water withdrawal* ***(water demand plus losses)****, by using empirical ratios of* ***irrigation efficiency (ratio of ET increase to water withdrawal)*** *or specific rules according to the irrigation* ***technique*** *(Rost et al., 2008, Jägermeyr et al., 2015).*

We propose to highlight the different types of shortcoming and flaws in the modelling framework, and its effects on irrigation efficiency, in discussion (section 6 of the original manuscript), as **described in the answer to observation 13, reviewer 1**.

We propose to clarify what is included in LSMs (return flows) and what is not included (conveyance losses) by changing the current sentence, so it is clearer (line 56 of original manuscript):

*Some LSMs prescribe irrigation rates estimated offline (Lo and Famiglietti, 2013; Cook et al., 2015), but most of LSMs and some GHMs estimate irrigation demand by calculating a deficit, for instance, a soil moisture deficit between actual and a target soil moisture (Haddeland et al., 2006; Leng et al., 2014; Pokhrel et al., 2015; Jägermeyr et al., 2015). **Some LSMs, which benefit from a physically based description of surface runoff and drainage, can explicitly calculate return flow**, but **conveyance** losses are not explicitly included (Yin et al., 2020; Leng et al., 2017).*

We propose to add a point at the end of section 4. Sensitivity analysis and parameter tuning, so it is clear that our modelling framework lacks some important characteristics of irrigation that could affect water withdrawal, ET increase and ultimately irrigation efficiency, in line 334 of the original submitted paper:

*After this analysis we underline four points. First, this process does not correspond to a proper calibration, as we assumed uniform parameter values, the number of simulations is low and the observed data is sparse. The objective of the sensitivity analysis and parameter tuning was to identify key parameters and reduce the underestimation of irrigation by tuning the uniform parameter values. **Second, our scheme does not include conveyance losses although application losses and return flows are represented. As ORCHIDEE determines the water partitioning, some model flaws in hydrologic processes like infiltration or bare soil evaporation could bias the effect in return flows, in the increase of ET, and ultimately in the irrigation efficiency.***

4.  (4) Could you add a description about the crop calendar? It should be explained how the authors defined the irrigation period. The authors mention that they did not consider double cropping, but there does not seem to be any explanation about the crop calendar in the current manuscript.

We propose to include the information on crop calendars after line 102 of the original manuscript:

*(...) so that C3 and C4 crops are simply assumed to have the same phenology as natural grasslands, but with higher carboxylation rates and adapted maximum possible LAI (Krinner et al., 2005). **The crop growing season depends on mean annual air temperature, as detailed in  (Krinner et al., 2005). In cool regions it starts after a predefined number of growing degree days, while in warm regions, it starts a predefined number of days after soil moisture has reached its minimum during the dry season. In intermediate zones, the two criteria have to be fulfilled. The end of the growing season also depends on temperature and water stress, and on leaf age.***

We propose to include the differences in crop calendar and growing season as a source leading to differences in irrigation amount, efficiency and irrigation dynamics (section 6 of the original manuscript), as **described in the answer to observation 13, reviewer 1**.

5.  (5) How does ORCHIDEE define "renewable"-groundwater resource?

It corresponds to shallow aquifers recharged by drainage at the soil bottom. We include this information in the manuscript, see response to observation 9 reviewer 1.

6. (6) All figures are blurred. It seems that dpi needs to be higher.

We propose to explore this problem with the edition staff. Currently we are using 350 dpi, but we can increase the resolution of our figures if needed.

7. (7) Regarding Fig4b-c, could you provide (supplementary) figures in % compared to Sackes et al. 2009? It is difficult to see how the irrigation bias is critical compared to the reference values.

We propose to include the difference in % in the supplementary as Fig. S11. We also propose to include a map of irrigation efficiency (ratio of ET increase to water withdrawal) by country.

We also propose to include the following text in the supplementary:

**In Fig. S11-a we can observe a strong overestimation (red). These areas depict a small irrigation rate (0.01 to 0.05 mm/d) that is strongly surpassed by the simulations, but the absolute value remains small. On the other hand, we observe areas with a strong underestimation (blue). These areas show higher irrigation rates than the areas in red (over 0.1) and in general, fit well with regions where paddy rice is important.**

**Irrigation efficiency map by country (in Fig. S11-b) show values over 100% in some countries. These high irrigation efficiency values mean that the crops increase ET by using a higher fraction of rainfall, even when there is not irrigation in the area. This is the result of suppressing part of the crop water stress, and lacking a specialized phenology module with crop stages like germination and harvesting. As crops are not harvested, even if there is not irrigation, there is more ET.**

[Figure]

a) Difference on irrigation, Irr - Sacks et al., 2009, (1998-2002) , in %

b) Irrigation efficiency by country, in %

*Figure S11 Difference in % in water withdrawal between Irr (yearly average 1998-2002) and dataset from Sacks et al., 2009 (a). Irrigation efficiency from Irr simulation (yearly average 1998-2002) as the ratio of increase of evapotranspiration to irrigation withdrawal (b).*

We propose to cite this figure in line 347 of the original manuscript, and line 479 of the original manuscript:

*This could also be partially explained by the use of globally uniform values, as there could be important local differences on irrigation strategies within the same country, and it remarks the need to assess the irrigation bias at different scales (See Fig. S11).*

*When we compare the simulations with the FLUXCOM product, the activation of irrigation leads to a reduction of the negative evapotranspiration bias, but the use of a single soil column in ORCHIDEE for both rainfed and irrigated crops could induce an overestimation of ET increase (See Fig. S11, in some cases the irrigation efficiency by country is too high).*

< Minor comments >

8. Line 43: PET needs to be spelled out here.

We propose to change this sentence, please see response to observation 3 and 4 from reviewer 1.

9. Line 44: If I remember correctly, H08 applies the soil moisture target method.

The reviewer is right, we will retire 'Hanasaki et al., 2018' from this citation, as H08 applies the soil moisture target method.

10. Line 60: In this context, GHM should also be included.

We agree, we propose to slightly change the sentence:

*In addition, irrigation shortage due to water availability is not well represented in **those** LSMs **(and GHMs) including this feature**, as some of them include a virtual infinite reservoir to fulfill irrigation demand (Ozdogan et al., 2010; Leng et al., 2014; Pokhrel et al., 2012).*

11. Table1: Probably, Ai should be ai.

Thanks for this typo, we will correct it

12. Line 271: It would be better to explain the original spatial resolution of the observed data in Section 3.2.

We will add the lacking resolutions for FLUXCOM, GLEAM and LAI3g

*We use two datasets: the first product is GLEAM v3.3a, which combines satellite-observed values of soil moisture, vegetation optical depth, and snow-water equivalent, reanalysis of air temperature and radiation, and a multisource precipitation product **at 0.25º of gridcell size** (Martens et al., 2017). The second dataset is FLUXCOM (Jung et al.,2019), which merges Fluxnet eddy covariance towers with remote sensing (RS) and meteorological (METEO) data using machine learning algorithms **at 0.5º of gridcell size**. Here we use RS+METEO products, specifically the averages of RS+METEO$_{WFDEI}$ and RS+METEO$_{cruncep,v8}$, to cover the analysis period.*

*We use the LAI3g dataseT (Zhu et al., 2013) climatological values for the period 1983-2015. This dataset applies a neural network algorithm on satellite observations of the Normalized Difference Vegetation Index (NDVI) 3g to estimate LAI **at 1/12 degrees of spatial resolution**.*

13. Fig5a: Could it be possible to add a reference plot(s) (AQUASTAT or other models' estimate)?

We propose to add a dashed line showing the AQUASTAT estimate for the year 2000.

14. Line 407, "we observe higher peaks and low values In Huang He when irrigation is activated": I can not understand which "low values" is about. Could you rephrase this?

Thanks for this observation. We propose to reformulate as follows:

*We now focus on the average TWSA value at the basin scale (Fig. 7). Activation of irrigation induces small changes in TWSA, which is coherent with changes in TWS between both simulations (Figure S2). For instance, we observe higher peaks in the TWSA values **in Huang He when irrigation is activated. Low values also become lower for the Irr simulation in Huang He basin.** In the Ganges river basin, low values are lower **as well** in the Irr simulation than in NoIrr.*

15. Line 418-420: I could not understand the point of this sentence in my first reading. Could you exemplify basins in Fig8 in this sentence?

We propose to put some examples, as suggested by the reviewer:

*The main effect of irrigation over the seasonal variations is that peak discharge can occurs before in the Irr simulation **(for instance Missouri river or Yellow river)**, or that the decrease after the peak is more rapid and low values are lower in Irr than in NoIrr **(for instance Colorado river or the Danube river)**.*

16. Fig 6: Add x-axis label.

We propose to add the x-axis label as 'Irrigation classes'.

17. Line 452: Hanasaki et al. 2008b seems to be a wrong reference here because H08 uses groundwater resource when surface water availability is not sufficient to meet irrigation demand.

We disagree with the reviewer. In Hanasaki et al., 2008, in simulation IRG they use a single imaginary water source. In FUL simulation, they make the hypothesis that all the water withdrawn comes from the rivers. In Hanasaki et al., 2018, the model includes a new groundwater recharge, and allows groundwater abstraction (Section 2.1, Newly added schemes). We propose to leave the citation as it is.

18. Line 459, "… like topography and environmental flow": Refer Hanasaki et al. 2018 here.

We will refer to Hanasaki et al. 2018 in this line.

19. Line 461: The following models also include detailed irrigation schemes: doi:10.5194/hess-19-3073-2015, doi:10.1029/2022MS003074.

We thank the reviewer for this paper, we propose to include the reference in our paper, line 460 for Yao et al., 2022. The reference Jägermeyr et al., 2015 has been already cited in the manuscript.

*Finally, the conditions to trigger irrigation, although controlled by four parameters, may seem too simple in our scheme, especially compared to specialized irrigation models, **the new irrigation scheme in LSM CLM5 (Yao et al., 2022), or the ISBA LSM (Druel et al., 2022),** which implement complex sets of rules to represent different irrigation strategies.*

**Final note:**

We also corrected the style of some sentences, so they were more understandable, and some orthographic and grammar errors.

[revised manuscript text omitted]

---

## Author Response (AR2)

**Response to Reviewers' observations on manuscript "Validation of a new global irrigation scheme in the land surface model ORCHIDEE v2.2"**

Pedro F. Arboleda-Obando, Agnès Ducharne, Zun Yin, Philippe Ciais

Correspondence: Pedro F. Arboleda-Obando (pedro.arboleda_obando@upmc.fr)

**Reply to topic editor**

Reviewer 1, in their re-review, did not feel that the additional analyses and discussion sufficiently addressed their concerns. However, I feel that the additions go a long way toward improving the manuscript to a point where it can be published after minor revisions. Some requests based on Reviewer 1's re-review:

We thank the topic editor and the anonymous reviewers for the time they spent reading and commenting on our paper.

1.) Please add discussion of how negative biases caused by the (in Reviewer 1's words) "missing irrigation between sowing and greening (defined by the threshold LAI)" could be addressed in future model developments.

We added a discussion on the possibilities to address the missing irrigation between sowing and greening, and in general, on rules to trigger and stop irrigation (after line 511 of the submitted manuscript, in bold the changes):

*Some rules could change the moment when irrigation is triggered and increase the amount (for instance allowing irrigation some days before the crop emergence) or decrease **it** (for instance, preventing irrigation during maturity of the crop, **shortening the growing season**, or preventing continuous irrigation during more than a certain number of days). **Implementing these sets of rules for irrigation strategies in ORCHIDEE is feasible, for instance the definition of the growing season (with trigger of irrigation before sowing and stop before harvesting) could be based on the prescription of start and ending dates as done by Yin et al., 2020, or could use the phenology information simulated by the model (as in the version used here, or using a crop-specific module as in Wu et al., 2016). But defining the set of rules and parameter values would need a careful tuning and evaluation process, with local data at sub-yearly scale.***

2.) Please discuss how the global parameterization work performed here can fail to address regional errors caused by missing processes etc. This should be tied in with the Discussion text added about the model's limitations.

We discussed how some of the limitations listed at the beginning of discussion could have an impact on regional errors observed in results. We did so by using some of the observations and ideas from reviewer 1. We also added a general view on how to improve these errors. We added these paragraph after line 495 of the submitted manuscript, so it is directly related to the list of model's limitations:

**These shortcomings and limitations could induce positive or negative biases in the simulated regional irrigation amounts; this as a result of differences in regional landscape, hydro-climatic conditions and local irrigation practices not well represented or absent in our scheme. For example, the missing representation of paddy irrigation induces under-irrigation in paddy rice areas, the joint representation of rainfed and irrigated crops induces over-irrigation in areas with other crop types and irrigation techniques, and the simplistic parameter tuning could tend to minimize the overall net bias, while increasing regional biases. These limitations (some shared with other global LSMs) call for further model developments that aim at a better representation of the water supply (fossil groundwater and water adduction to list two mentioned in the results) and the water demand (a separate water budget for irrigated areas, the inclusion of other irrigation techniques, new irrigation rules such as irrigation before sowing or interruption of irrigation before harvest). In addition to the improvements noted here that focus on model developments, the irrigation representation can be improved by using new input datasets and regional parameter values to include local practices (if these datasets exist at the coarse model resolution in the global domain, and for historical period or future scenarios). For instance, to prescribe regional $\beta$ values, or to prescribe the start and end of the growing season.**

3.) Please revise figures and equations as suggested by Reviewer 1. Please also review figures and equations not mentioned for similar issues.

We revised figures and equations. Please see responses to reviewer 1 below.

4.) Please convert all tables in the Supplement to text, rather than images. This is important for readability and accessibility.

We converted the tables in the Supplementary to text.

**Review comments on 'Validation of a new global irrigation scheme in the ORCHIDEE land surface model' by Arboleda-Obando et al.**

I was the Reviewer #1 in the previous review round. The authors agreed with the major technical flaws in the proposed irrigation scheme in ORCHIDEE I pointed in the previous review. The flaws can cause either positive (sub-grid-scale parameterization scheme) or negative (missing irrigation between sowing and greening stages) biases in irrigation. The authors correctly acknowledged the issues the flaws can cause and added additional discussions and supporting results (e.g., Figure S9) that demonstrate it.

We thank the anonymous reviewers for the time he spent reading and commenting on our paper.

However, the revised manuscript makes no effort of attempting (or at least proposing solutions) to address the flaws other than just acknowledging the issues. It is understandable that some of the deficiencies in the proposed irrigation scheme cannot be readily fixed for the manuscript, but the authors should be able to propose ways to improve them. For example, missing irrigation between sowing and greening (defined by the threshold LAI) can be easily added because ORCHIDEE already considers phenology information even if the model used in this work is not the version of Wu et al. (2016).

We thank the reviewer for this observation. We added a paragraph that partially explains some ways to improve the irrigation limitations (see response to observation 3 from the editor) and we add some specific ideas for the missing irrigation between sowing and greening (see response to observation 2 from the editor).

Another major issue relevant to above is that the authors do not link the apparent specifics of output errors to the deficiency in the current model. Instead, potential error factors in irrigation scheme, parameter tuning, and forcing data are listed in isolation from the result in the beginning of the Discussion. For example, while the added discussion and figures (see Figure S9) show positive bias in irrigation (over-irrigation) likely caused by irrigating the whole grid cell that contains a small fraction of the cell, the bias becomes negative gradually with increasing irrigated cropping fraction. It can be interpreted that, while over-irrigating fractional 'irr cells' causes +ve bias, late-start of irrigation (LAI based thresholding) may cause overall -ve bias (under-irrigation), with their net bias minimized by tuning beta. If this is the case, minimizing the global irrigation bias caused by a combination of systematic biases can explain the large regional irrigation biases. Current discussion does not provide useful insights to improve the specific issues reported in the manuscript.

We thank the reviewer for this observation. We felt that clearly stating the flaws of our approach was sufficient to enlighten the ways to the representation, and we apologize for this misjudgement. We added a sentence linking output errors and model deficiencies based on these observations from reviewer 2, and added a general view on the ways to address some of these flaws (see response to observation 3 from the editor).

As a result, although the manuscript reports some improvement in ET bias in comparison with FLUXCOM and small improvement in TWSA against GRACE, overall efficacy of the proposed irrigation scheme is not very convincing, particularly at regional scales. I recommend that the manuscript can be considered for publication after addressing comments listed above and convincingly explain specifics of errors with quantitative tests rather than generic speculations.

• Comment: Figure 9 inset text: Reduce the font size to make the whole text visible

• Reply: We corrected a small typo in the inset text, but we do not fully understand which text is not visible in this figure. We propose to discuss with the editor to correct this observation.

Part of the inset text '-633.9 km3/year' is still trimmed by the subfigure frame and the inset text is distinctively unbalanced in size in comparison with the other subfigures in the same figure. In fact, overall quality control of figures in the main text and supporting material is in poor quality and not acceptable for published articles. I will list just some of them below.

We thank the reviewer for clarifying this observation. We balanced the size of the inset text, and we reduced the font size of the subfigure. We also corrected the superscript units.

• Figure 2: 3 in km3 and 2 in m2 should be in superscript

We corrected the superscript units.

• Figure 3: km3

We corrected the superscript units.

• Figure 4: km 3 and inconsistent use of 'y', 'year', and 'Year' in unit

We corrected the superscript units and the inconsistent use of 'year'.

• Figure 5: Consistency between mm/day or mm/d throughout the manuscript

We corrected the inconsistent use of mm/day to mm/d in the figures to be consistent with the manuscript.

• Figure 6: m2

We corrected the superscript units.

• Figure 7: Incorrect y axis label (TWSA)

We added TWSA to the y axis label so the variable is clearly stated.

• Figure 8: Missing y axis label

We added a y axis label, with variable and units

• Figure 9: gap between d) and subfigure title; inconsistent font size of subfigure titles and color index tick labels; inset texts font size

We corrected the gap between d) and the subfigure title; we set the font of the subfigures to the same size. We set the tick labels size and the inset text font size.

• Equations 1-2 in Supp: Period (.) is not for 'product' operator

We are sorry for this typo. We changed the Period operator by the product ($\times$) operator

• Figure S1: km3

We corrected the 3 and set it as superscript.

• Figure S2: m^2; missing units in the subfigure title; (c ) and (d ) in the figure caption; too large font size of subfigures

We corrected m^2 and put it as superscript. We added figures and time steps to the subfigures title, and reduced the font size. We corrected figure captions c and d. We corrected the 3 and set it as superscript.

• Figure S3: m^2 and missing 'degree' symbol in temperature unit; too large font size of subfigures

We corrected the ^2 as superscript, added the degree symbol and reduced the font size of the subfigures. We added figures and time steps to the subfigures title.

• Figure S4: too large font size of subfigures

We set the smaller font size of subfigures

• Figure S5: inconsistent font size in inset and color bar

We corrected the inconsistencies in font size in the color bar, we deleted one of the color bars as they were the same for both figures.

• Similar errors in the subsequent figures (I will stop here)

We corrected superscript errors and the subfigures font size inconsistencies in the subsequent figures. We thank the reviewer for taking the time to point out these errors and inconsistencies.

• Table S1: poor-quality screenshot?

The table was converted to text, and it was set to fit the page size.

• Table S2: a little bit better quality but still a screenshot?

The table was converted to text, and it was set to fit the page size.

• Inconsistent fonts between tables

Font size was set to fit the document, and inconsistencies were corrected.